# Intrinsic Credit Assignment for Long Horizon Interaction

Ilze Amanda Auzina [* 1 2]   Joschka Strüber [* 1 2]   Sergio Hernández-Gutiérrez [* 1 2]
Shashwat Goel [† 3 4]   Ameya Prabhu [† 1 2]   Matthias Bethge [† 1 2]

🌐 bethgelab.github.io/delta-belief-rl   ⭘ delta-belief-rl

## Abstract

How can we train agents to navigate uncertainty over long horizons? In this work, we propose $\Delta$Belief-RL, which leverages a language model's own intrinsic beliefs to reward intermediate progress. Our method utilizes the change in the probability an agent assigns to the target solution for credit assignment. By training on synthetic interaction data, $\Delta$Belief-RL teaches information-seeking capabilities that consistently outperform purely outcome-based rewards for RL, with improvements generalizing to out-of-distribution applications ranging from customer service to personalization. Notably, the performance continues to improve as we scale test-time interactions beyond the training horizon, with interaction-efficiency increasing even on Pass@k metrics. Overall, our work introduces a scalable training strategy for navigating uncertainty over a long-horizon, by enabling credit assignment to intermediate actions via intrinsic $\Delta$Belief rewards.

## 1. Introduction

In 2025, language agents have rapidly improved in their ability to solve fully specified tasks, ranging from challenging questions posed by researchers (Schmidgall et al., 2025) to economically valuable tasks (Kwa et al., 2026; Patwardhan et al., 2025). However, they still struggle with under-specified or open-ended problems, which require interactions to seek information before a solution can be attempted (Grand et al., 2026). Whether it be navigating the hypothesis space in a scientific problem, or interacting to understand user's preferences, models struggle to ask the right questions (Laban et al., 2025).

---
[*]Equal contribution, [†] Equal supervision [1]Tübingen AI Center [2]University of Tübingen [3]ELLIS Institute Tübingen [4]Max Planck Institute for Intelligent Systems. Correspondence to: Ilze Amanda Auzina <ilze.auzina@bethgelab.org>.

*Proceedings of the $43^{rd}$ International Conference on Machine Learning*, Seoul, South Korea. PMLR 306, 2026. Copyright 2026 by the author(s).

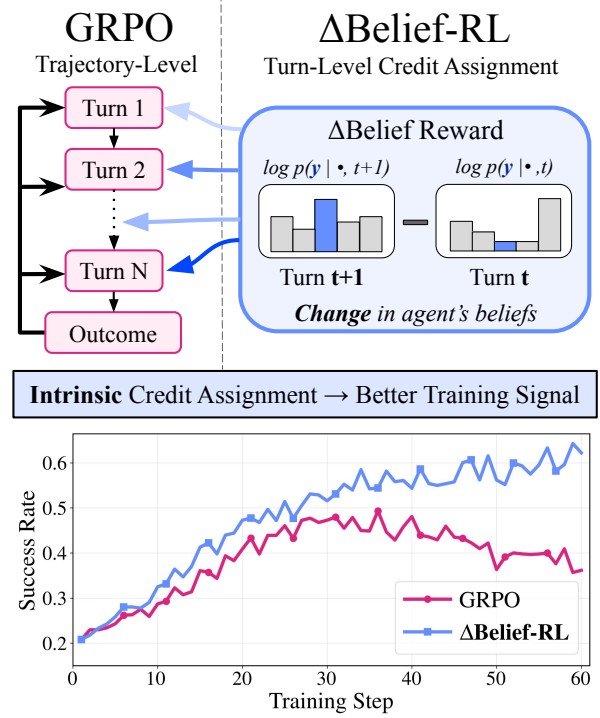

*Figure 1.* **Main contributions.** We propose a dense reward signal, $\Delta$Belief Reward, based on agent intrinsic belief updates in long horizon tasks. We find that **(1)** $\Delta$Belief-RL leads to more sample-efficient training **(2)** our trained agent generalizes better to unseen information-seeking tasks; and **(3)** scales better with increased test-time interaction budget.

The root of this disconnect lies in the pre-training data itself: web data does not fully capture information-seeking. Although the web provides a vast repository of user-posed questions and answers, this data rarely reflects the underlying internal beliefs or knowledge gaps that prompted the user to ask the question itself. The utility of a question is inherently tied to an agent's **own** beliefs: a useful question for one agent may not provide new information to another with different knowledge and uncertainty distributions.

In this work, we study how agents can be trained to navigate uncertainty across long-horizon interactions. Such trajectories require granular credit assignment to distinguish which intermediate actions drive success versus those that lead to failure. A common strategy in RL is to use a learned critic

to estimate a *value function* (Sutton & Barto, 2018), but this approach is often prohibitively expensive for large-scale language models. Consequently, current approaches use only sparse outcome rewards (Guo et al., 2025). The hope is that, while sparse rewards lack granular credit assignment, the agent will eventually converge towards a globally optimal solution. While this simplification holds for single-turn or fully specified tasks, it is ill-suited for open-ended, partial-information settings that require the model to actively explore and seek information at *every step*.

We propose a scalable solution, ΔBelief-RL, a novel framework which guides learning and credit assignment on intermediate actions in long-horizon tasks. At each interaction during training, we monitor the change in agent's belief towards the target. We use this information as a dense training signal, ΔBelief reward, that reinforces actions which shift the agent's internal beliefs toward the target, while also using the final outcome as part of the reward. The benefit of our method is that it does not require a separately trained critic or process reward model, but rather uses the agent's own intrinsic beliefs as a proxy for each action's value. This enables intermediate credit assignment "for free" by leveraging the agent's own progress toward the solution. Furthermore, our training strategy is general-purpose; it can be applied to any task where the correct final outcome is available during training.

We evaluate ΔBelief-RL by training Qwen3 models (up to 4B parameters) on a classic long-horizon information-seeking task: 20 Questions. We name the resulting models CIA, Curious Information-seeking Agent (CIA), optimized for active inquiry. Our approach consistently outperforms standard, sequence-level GRPO training, such as STARPO (Wang et al., 2025), across all model sizes (Table 1). Notably, our CIA-1.7B model outperforms DeepSeek v3.2 (670B), supporting our hypothesis that specialized training is essential for active information-seeking where web-scale pretraining and RLVR fall short.

Remarkably, the obtained capabilities do not plateau at the 20-turn training cap; instead, performance continues to scale as test-time interactions extend beyond the training horizon (Figure 8). Furthermore, ΔBelief-RL expands the agent's reasoning capabilities beyond its initial training (Figure 6). Crucially, these information-seeking behaviors generalize to novel tasks not seen during training (Figure 9, Figure 10), demonstrating that ΔBelief-RL imparts task-agnostic ability to efficiently interact and navigate uncertainty.

Overall, we show how an agent's internal beliefs can act as an intrinsic reward signal for credit assignment in uncertain long-horizon tasks. This yields substantial gains in interaction-efficient information-acquisition, an important component of intelligence (Chollet, 2019) and world modeling (Warrier et al., 2025). We anticipate that as agents

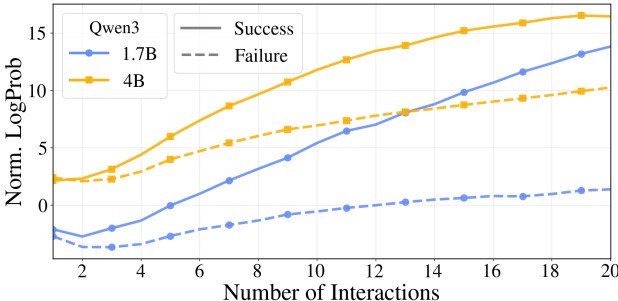

*Figure 2.* **Belief updates.** The per-turn beliefs of about the ground-truth Qwen3 (1.7B and 4B), split by final outcome. The trajectories were generated by DeepSeek-v3.2. On average, beliefs steadily increase and the rate of growth strongly correlates with the outcome of the trajectory.

develop more sophisticated implicit world models, their belief updates will become even more precise, further amplifying the effectiveness of our approach. Ultimately, we hope this provides a path toward agents that can reason about their own uncertainty to efficiently pursue open-ended goals.

## 2. Background and Problem Setup

We consider a multi-turn interaction setting where an agent must uncover a latent target concept $y \in \mathcal{Y}$. Each episode begins with a task-specific prompt that defines the search space. To solve the task, the agent interacts with the environment over a long horizon, generating a trajectory $\tau$ of $N$ steps: $\tau = \{(a_1, o_1), \ldots, (a_N, o_N)\}$, where $a_t$ represents the agent's action (e.g., a query or hypothesis) and $o_t$ represents the environment's response (e.g. user's answer) at time $t$. In our setup, both the actions and the observations are strings. We denote the interaction history up to step t as $h_t = \{a_1, o_1, \ldots, a_{t-1}, o_{t-1}\}$, where the agent samples each action based on the accumulated context: $a_t \sim \pi_\theta(\cdot \mid h_t)$. The initial prompt can be considered a part of $h_1$. At the end of the trajectory $\tau$, the environment returns a verifiable score $r(\tau)$ which assesses whether the agent successfully uncovered the target concept $y$. We define the policy's performance over the concept space $\mathcal{Y}$ as $\mathrm{Perf}(\pi) = \frac{1}{|\mathcal{Y}|} \sum_{y \in \mathcal{Y}} \mathbb{E}_{\tau \sim \pi}[r(\tau)]$. We train on a finite subset $\mathcal{Y}_{\text{train}}$ and evaluate generalization on unseen instances $\mathcal{Y}_{\text{test}}$.

### 2.1. Training Environment

We use the 20 Questions (20Qs) game as our training environment for multi-turn interaction. In this setting, the target concept $y$ is a "secret word" (e.g. "belief"). The agent's actions $a_t$ are yes/no inquiries (e.g. "Is the word an abstract concept?"). The observations $o_t$ are responses provided by a user simulator (e.g. "Yes"). The agent's objective is to identify $y$ within a maximum of 20 turns. An episode terminates successfully if the agent correctly guesses $y$, at which point it receives the outcome

reward $r(\tau)$; otherwise, the interaction concludes after 20 turns with a score of zero. This setup is representative of a broad class of information-seeking problems where agents must optimize interactions to arrive at a target solution.

**Agent Configurations** We instantiate the 20 Questions environment with the training policy (`Qwen3`) at two scales, 1.7B and 4B, as the questioner agent, and `Qwen3-14B` as the *User simulator*. To fit full trajectories within the context window for backpropagation, we use the models in non-thinking mode.

## 3. $\Delta$Belief-RL: Intrinsic Credit Assignment

The $\Delta$Belief-RL framework leverages an agent's internal beliefs as an intrinsic reward signal, enabling dense credit assignment in probabilistic, long-horizon tasks. In the context of 20 Questions, this involves quantifying the shift in the agent's posterior distribution over the target concept $y$ following each interaction $(a_t, o_t)$.

The following sections detail our methodology. First, we formalize the measurement of beliefs for language model based agents. Then, we define a metric to capture progress toward the goal at each turn, and empirically validate that it correlates with task success. Finally, we describe how we utilize these intrinsic signals to optimize the agent's information-seeking capability with RL.

**Agent Beliefs** We elicit the agent's internal belief by leveraging its underlying token probability distribution. Given an elicitation prompt $e_i$, we calculate the belief at turn $t$ as the probability the policy assigns to the target concept $y$ conditioned on the history $h_t$: $b_t = p_\theta(y_i \mid h_t, e_i)$. For 20 questions, we use $e_i =$ "Is the secret <target>". For prompt design details see App. A.1.1.

**$\Delta$Belief reward: Belief Change Signal** By tracking $b_t$ across the trajectory, we can quantify how each interaction resolves uncertainty, shifting the model's internal "world view" towards the correct solution. We calculate the per turn belief-change, denoted $\Delta$Belief, as the log-ratio of sequential beliefs:

$$\Delta\text{Belief}_t = \log \frac{b_t}{b_{t-1}} = \log b_t - \log b_{t-1} \qquad (1)$$

By utilizing log-probabilities, we ensure numerical stability and prevent floating-point underflow during training. This dense, turn-level signal reinforces actions that lead to the most informative updates to the agent's internal world view.

### 3.1. Validating the $\Delta$Belief Measurement

The effectiveness of the $\Delta$Belief reward depends on the agent's internal world model making sufficiently calibrated

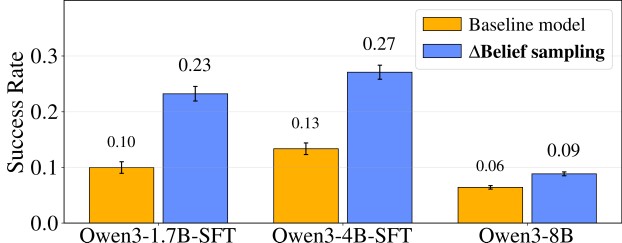

*Figure 3.* **Best-of-8 sampling with $\Delta$Belief.** Success rate on the 20 Questions task for our baseline models `Qwen3-1.7B` and `Qwen3-4B` after SFT, as well as the base model `Qwen3-8B`. We compare regular question generation with $\Delta$Belief sampling: we sample 8 questions at every turn and select the one that maximizes $\Delta$Belief. Across sizes, we observe a significant rise in performance when our signal is employed to guide the sampling of questions.

updates based on environment feedback. We confirm this using two experiments: first, we check whether the belief assigned to the correct answer reliably correlates with the trajectory outcome, and the number of information-seeking actions; second, we evaluate whether action selection based on $\Delta$Belief at inference-time leads to performance gains.

**Do Agent Beliefs Reflect Interactive Progress?** To determine whether internal beliefs reliably track environmental information, we decouple the belief update process from the question-asking policy to test it in isolation. The belief updates of `Qwen3` on off-policy trajectories generated by `DeepSeek-v3.2` are shown in Figure 2. We find that beliefs consistently increase over time, confirming that these models successfully integrate evidence from observations. Crucially, these updates correlate with task success: successful episodes exhibit significantly higher cumulative belief increases after few turns. This suggests that the agent's actions effectively narrow the solution space, moving its internal "world view" closer to the target concept.

**Does Optimizing Belief Updates Improve Task Success?** Next, we examine whether optimizing for $\Delta$Belief reward leads to more successful interactions compared to the baseline SFT policy (details Sec. 4). We perform an intervention with best-of-n sampling to simulate an agent that explicitly optimizes for belief updates during inference. At each turn $t$, we sample $n = 8$ candidate questions, simulate the environment's response for each, and select the action that maximizes the immediate belief change:

$$a_t \leftarrow \arg\max_{k \in \{1,\ldots,n\}} \Delta\text{Belief}(a_{t,k}) \qquad (2)$$

As shown in Figure 3, maximizing belief updates significantly and consistently improves performance across all model scales. While this "look-ahead" method is unsuitable for inference—as it requires oracle access to ground-truth answers during the selection process—it confirms that

ΔBelief reward is a powerful heuristic for exploration and a viable signal for RL training.

### 3.2. Training with Reinforcement Learning

We build upon the standard RL with verifiable rewards framework by augmenting the sparse outcome reward with our dense intrinsic signal at every turn $t$. Since the policy update occurs after a completed trajectory, we can backpropagate the final outcome to each individual step. We define the per-turn reward $r_t$ as:

$$r_t = \underbrace{r^{\text{eog}}}_{\substack{\text{trajectory} \\ \text{outcome}}} + \underbrace{\lambda \max(\Delta \text{Belief}_t, 0)}_{\text{intrinsic exploration}} + \underbrace{r_p}_{\substack{\text{efficiency} \\ \text{penalty}}} \quad (3)$$

where $\lambda$ scales the belief-based reward. The penalty term $r_p$ discourages wrong formatting and redundancy, while incorporating a number of turns penalty to encourage efficiency. Notably, we clip the intrinsic reward at zero, ensuring that agents are rewarded for increasing belief in the correct concept without being penalized for temporary decreases in confidence. We present ablations in Appendix A.1.

**Turn-wise GRPO**  While standard GRPO applies the same reward across the entire trajectory, this coarse approach obscures the specific contribution of individual actions. Since our framework provides a dense reward signal $r_t$ for every turn, we go beyond trajectory-level feedback and compute advantages at the *turn level*.

For a group of $G$ sampled trajectories $\{\tau^i\}_{i=1}^G$, let $r_t^i$ be the reward at turn $t$ for sample $i$. We calculate the group-relative turn-wise advantage as:

$$\widehat{A}_t^i = \frac{r_t^i - \text{mean}(\{r_t^i\}_{i=1}^G)}{\text{std}(\{r_t^i\}_{i=1}^G)}. \quad (4)$$

This advantage $\widehat{A}_t^i$ is assigned to all tokens generated in turn $t$. By replacing a uniform trajectory-level advantage with the sequence $(\widehat{A}_1^i, \ldots, \widehat{A}_N^i)$, we can provide the policy with granular feedback on each specific interaction. Although we instantiate this approach using multi-turn GRPO (Zeng et al., 2025), this turn-wise reward structure remains compatible with any standard reinforcement learning algorithm (details in App A.1.2, App A.1.3).

## 4. Experimental Details

**Data and Models**  We perform RL training on the 20Qs environment. The ground-truth concepts $\mathcal{Y}$ are split into four distinct sets: 341 for SFT training, 1,000 for RL training, 198 for validation, and 433 for testing. The set of possible secrets $\mathcal{Y}$ is kept unknown to the *Questioner* matching challenging open-ended settings. As mentioned in Section 2.1,

*Table 1.* **Success rate on the test set**. Average success rates and Pass@k of the SFT BASELINE, STARPO and CIA as well as two large API models on a held-out test set. Across both model sizes, CIA consistently outperforms all baselines.

| Method | Mean@8±std | Pass@8 |
|---|---|---|
| BASELINE (1.7B) | 9.97% ± 1.04% | 32.03% |
| STARPO | 16.54% ± 1.32% | 45.73% |
| CIA (ours) | **24.80%** ± 1.10% | **53.10%** |
| BASELINE (4B) | 13.34% ± 1.05% | 36.87% |
| STARPO | 24.36% ± 1.18% | 59.12% |
| CIA (ours) | **33.72%** ± 1.26% | **63.97%** |
| DeepSeek-V3.2 | 14.35% ± 0.87% | 47.34% |
| Qwen3-235B-Instr. | 8.83% ± 0.87% | 27.71% |

we select `Qwen3-1.7B` and `Qwen3-4B` as the agent and `Qwen3-14B` as the user simulator.

**RL Training.**  All RL experiments begin from a *Baseline (SFT)* checkpoint, which is fine-tuned on 8,918 demonstration turns generated via rejection sampling using GEMINI 2.0 FLASH. We train using turn-wise GRPO ($G$=16, temperature 1.0) with LoRA (rank 64, $\alpha$=64, LR $3 \times 10^{-5}$) on 2 NVIDIA H100 GPUs (one training the Questioner model, one for inference with the user simulator). The complete details are in Appendix E, F.

**Evaluation**  We compare ΔBelief-RL against the state of the art method for multi-turn settings, StarPO (Wang et al., 2025) which performs trajectory-level GRPO. We report results on a held-out 20 Questions test set, as well as out of distribution information-seeking tasks and benchmarks for two applications: customer service and user personalization.

## 5. Results: ΔBelief-RL Training Improves Interaction-Efficiency

**ΔBelief-RL evokes information-seeking.**  Our framework, ΔBelief-RL, substantially improves information-seeking performance over both, standard GRPO and SFT, baselines. On the 20Qs test set, our agent, CIA, outperforms the SFT policy by **14.83%** and **20.38%** for the 1.7B and 4B models, respectively (Table 1). Relative to standard multi-turn GRPO (STARPO), it yields absolute improvements of **8.26%** and **9.36%**, indicating that the ΔBelief signal provides a meaningful intrinsic objective that improves credit assignment and, in turn, information gathering. These improvements also persist under higher pass@k evaluation, suggesting that the gains are not an artifact of a single sampled interaction. Finally, our trained CIA outperforms much larger inference-time baselines (`DeepSeek-V3.2` and `Qwen3-235B-A22B-Instruct-2507`) by **19.37%** and **24.89%** (4B model size), respectively, despite using ∼98% fewer parameters. The relatively low performance of

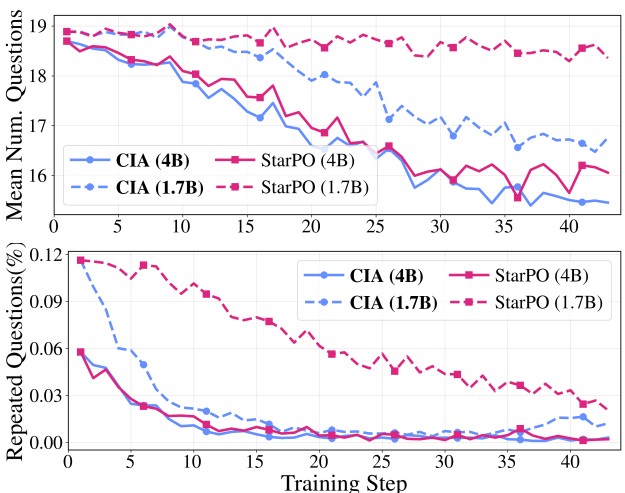

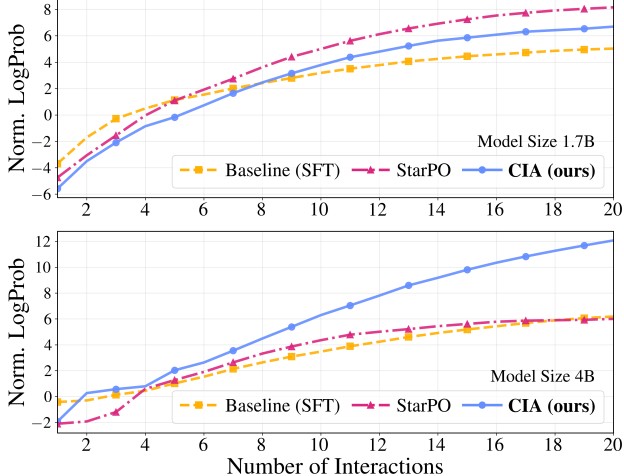

*Figure 4.* **Training dynamics.** For both, mean number of questions per episode (top) and mean fraction of repeated questions (bottom) during RL training, lower is better. Across both `Qwen3-1.7B` and `Qwen3-4B`, $\Delta$Belief-RL reduces the number of turns required to solve the game and suppresses redundant queries more rapidly than standard GRPO (STARPO).

*Figure 5.* **Belief-update dynamics.** Normalized elicited log-probability of the correct concept, $\log p_\theta(y_i \mid h_t, e_i)$, as a function of the number of interactions for 1.7B models (top) and 4B models (bottom). At the 4B scale, our method CIA shows the largest and most sustained increase in belief updates over multiple interactions, while STARPO remains close to the SFT baseline. For 1.7B models, both trained variants track the baseline closely.

these massive models further underscores the difficulty of our curated test set compared to prior benchmarks.

**$\Delta$Belief reward promotes efficient exploration.** While both standard-GRPO and $\Delta$Belief-RL incorporate an explicit per-turn efficient exploration penalty (see Section 3.2), their exploratory behaviors differ significantly. Our results show that $\Delta$Belief-RL yields a faster reduction in both repeated inquiries and total questions per episode (Fig. 4). This suggests that the $\Delta$Belief reward does not merely reinforce success, but actively encourages higher-information queries. By maximizing belief updates, the agent learns to resolve uncertainty with fewer interactions, beyond what a simple trajectory-length penalty can achieve.

**$\Delta$Belief-RL leads to faster belief updates.** We investigate whether our training framework significantly shifts the belief update patterns compared to the initial SFT policy. On the 4B scale, $\Delta$Belief-RL (CIA) demonstrates the largest and most sustained increase in belief updates between interactions, while STARPO remains closely aligned with the baseline policy. In contrast, no observable differences appear on the 1.7B scale (Fig. 5). These results suggest that our framework is particularly effective for models with more calibrated internal world models, as it results in more pronounced belief shifts.

**$\Delta$Belief-RL improves pass@k success rates at high k.** A common critique of RL post-training in language models is its tendency toward 'policy sharpening,' wherein the model primarily increases pass@1 performance, upweighing existing successful trajectories. To determine whether $\Delta$Belief-RL also improves exploration, we perform a low-

variance pass@k analysis (Chen et al., 2021). Our results show that CIA maintains a consistent advantage over the SFT baseline policy even at high k values (up to 128) (Fig. 6). Our intrinsic credit assignment strategy reinforces good actions, penalizing bad ones, lead to compounding benefits over long trajectories.

### 5.1. Ablations

**Generalization to stronger user simulators.** Real-world deployments require agents to interact with a wide range of users whose response distributions differ from the training environment. To ensure that our agent does not overfit to the training simulator, we evaluate it to interact with a diverse set of stronger answerers. We swap the training answerer (`Qwen3-14B`) for significantly larger API models at test time and disable auxiliary software verifications. We find that the success rates remain robust for all these changes (Table 2). This confirms that our training does not rely on specific artifacts of the training-phase user simulator.

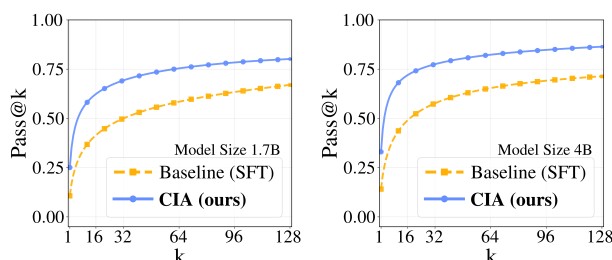

*Figure 6.* **Pass@k on Twenty Questions.** Pass@k as a function of the number of sampled trajectories $k$ for 1.7B models (left) and 4B models (right). Our method, CIA, consistently achieves higher pass@k than the Baseline across the entire range of $k$ (up to 128).

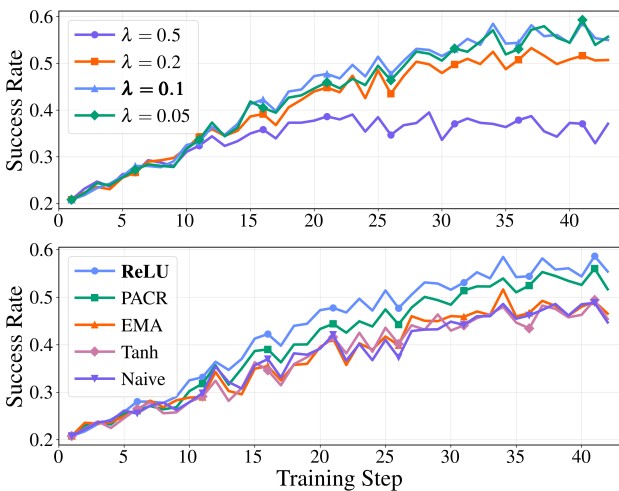

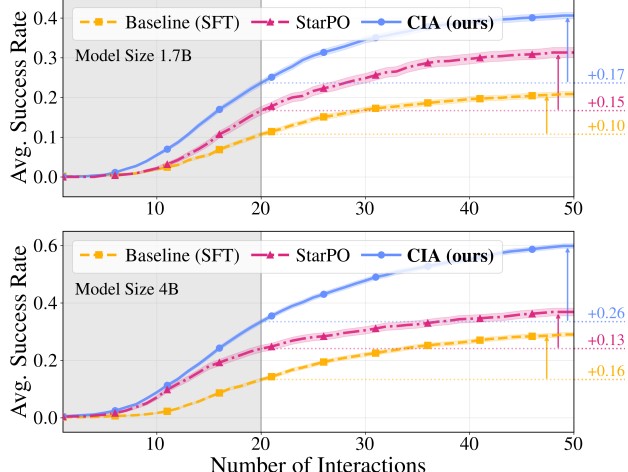

*Figure 7.* **Training Ablations.** Top: Ablation of the reward weight $\lambda$. Moderate values ($\lambda \in \{0.05, 0.1\}$) optimize learning progress, whereas excessive weighting leads to suboptimal convergence. Bottom: Ablation of $\Delta$Belief Reward normalization. Among normalization methods, non-negative shaping (ReLU), provides the most stable and highest-performing training dynamics.

*Figure 8.* **Test-time interaction scaling.** Average success rate as a function of the interaction budget, increased up to 50 turns, for 1.7B models (top) and 4B models (bottom). Our model, CIA, exhibits larger performance gains within the standard 20-interaction regime and continues to improve as the interaction budget is extended than both STARPO and the Baseline SFT policy.

**Balancing intrinsic and verifiable rewards.** We investigate the weighting term trade-off between the verifiable end-of-game reward and $\Delta$Belief reward. As shown in Fig. 7 (top), training collapses when $\lambda \geq 0.5$. While lower values ($\lambda = 0.05$) allow for stable learning, we find that a moderate weight of $\lambda = 0.1$ yields the best validation performance. Since $\Delta$Belief is unbounded while $r^{\text{eog}}$ remains within a fixed range, the optimal $\lambda$ serves to align the magnitudes of the two signals, ensuring a balanced training objective.

**Different normalization effects on $\Delta$Belief reward** As $\Delta$Belief reward is unbounded, appropriate normalization is crucial for stable optimization. We therefore compare several normalization schemes: *naive* (no normalization), *tanh* squashing, *ReLU* (positive-only shaping), *EMA* normalization using a lagged reference policy to reduce reward over-optimization (Hatamizadeh et al., 2026), and *PACR*, which restricts to the positive domain and applies min–max scaling (Yoon et al., 2025). As shown in Fig. 7 (bottom) right, naive,

EMA, and tanh lead to noticeably worse training dynamics. Naive and EMA introduce large-magnitude (often negative) updates that can dominate the verifiable reward and destabilize learning, while tanh overly compresses the signal and removes informative variation. In contrast, ReLU yields the most stable and highest-performing training. PACR performs comparably but slightly worse than ReLU, suggesting that restricting the intrinsic signal to positive rewards is beneficial, whereas additional min–max scaling can distort the relative informativeness of belief updates.

## 6. Generalization and Applications

In this section, we evaluate the generalization of CIA across three dimensions: (i) interaction scaling, assessing whether performance improves with additional inference-time turns (Sec. 6.1); (ii) task robustness, testing generalization to out-of-distribution benchmarks (Sec. 6.2); and (iii) practical transfer, evaluating performance on simulated environments designed to mirror real-world applications (Sec. 6.3).

### 6.1. Test-Time Interaction Scaling

**Success scales with increased interaction budget.** During training, episodes are strictly limited to 20 turns. Here, we investigate whether the agent's information-seeking capabilities generalize to longer interactions. By increasing the interaction budget to 50 turns in Fig. 8, we observe that CIA sustains large performance gains. This is especially pronounced at the 4B scale, where CIA achieves a STARPO 26% higher (absolute) success rate at the 50-turn mark. These results suggest that belief-based rewards do more than improve within-budget efficiency; they foster

*Table 2.* **Generalization to stronger user simulators.** Success rates (Mean@8) when evaluating agents with a different user simulator from the one used during training. Both the BASELINE and CIA are trained using `Qwen3-4B` as the base model and `Qwen3-14B` as the user simulator. Test-time performance remains similar on other user simulator models.

| Answerer | Baseline | CIA |
|---|---|---|
| `Qwen3-14B` (train) | 13.34% | 35.65% |
| `Qwen3-235B-A22B-Instr.` | 17.49% | 31.18% |
| `DeepSeek-v3.2` | 16.48% | 37.64% |

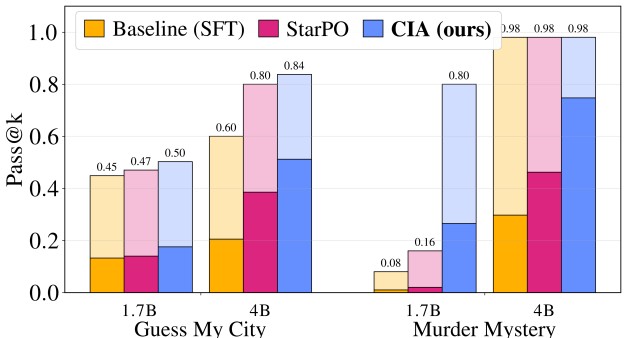

*Figure 9.* **OOD generalization.** Pass@k on two out-of-distribution benchmarks (*Guess My City* and *Murder Mystery*) Dark bars show Pass@1; the lighter stacked increment shows Pass@8.

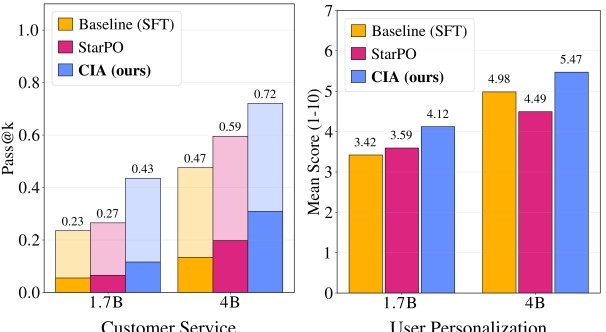

*Figure 10.* **Practical applications.** Mean score on the test-set for the OOD *User Personalization* and *Customer Service* benchmarks. For *User Personalization*, the sample-level raw scores are in the range [1, 10]. Across both sizes, CIA strongly outperforms the previous SoTA method, STARPO.

teach general information-seeking strategies that continue to resolve uncertainty as more evidence becomes available.

## 6.2. Out-of-Distribution Generalization

We evaluate out-of-distribution (OOD) generalization using three distinct tasks: *Guess My City* and *Murder Mystery* (Tajwar et al., 2025). Further details regarding environment configurations and their corresponding prompts are available in Appendices B and C, respectively.

**Generalization to held-out tasks.** Our CIA agent exhibits robust generalization, consistently surpassing SFT and STARPO baselines across all tested environments (Figure 9). These gains are most pronounced in the 4B CIA model, which maintains a significant performance lead over all competing methods.

On *Guess My City*, we observe a similar pass@1 advantage for CIA over STARPO (3.6% for 1.7B and 12.6% for 4B). Unlike the other benchmarks, the performance gap narrows at higher k values. This is likely due to the task's lower complexity; once the agent identifies the correct country, the remaining hypothesis space of cities is relatively small. Furthermore, the high SFT performance suggests this task's structure closely mirrors the 20-questions format of the training data. Qualitatively, CIA adopts a more efficient "top-down" strategy, asking abstract questions about city characteristics, whereas STARPO and SFT rely on simple, brute-force queries of specific locations (Appendix A.3.2).

The *Murder Mystery* task requires the evaluated model to solve a crime, having access to an oracle that can provide information upon request. On this task, CIA achieves its most significant gains over STARPO, with a lead of 24.5% (1.7B) and 28.5% (4B) at pass@1, as can be observed in Fig. 9. Performance at pass@k saturates quickly across all 4B models because the hypothesis space is inherently small, with only five suspects per task. Consequently, while pass@8 results are less discriminative due to the high probability

of a successful random guess, the pass@1 rate provides a more accurate reflection of the agent's reasoning quality. For detailed example rollouts, see Appendix A.3.4.

## 6.3. Generalization to Practical Applications

To further test the OOD generalization capabilities of our methodology, we evaluate CIA on settings that mirror practical applications more closely than the previously presented benchmarks. We evaluate CIA against the aforementioned SFT and STARPO baselines on the STaR-GATE dataset (Andukuri et al., 2024) (which we refer to as *User Personalization*), and on the *Customer Service* task (Tajwar et al., 2025). As aforementioned, further details regarding environment configurations and their corresponding prompts are available in Appendices B and C, respectively.

The *User Personalization* environment consists of multi-turn interactions between a user and the language model: the evaluated model must strategically elicit latent user preferences and background context through iterative questioning before delivering a final, informed response. Our results, summarized in Fig. 10, provide further evidence of CIA exhibiting enhanced information-seeking abilities across both model sizes, which translate into higher-quality user-alignment on this task; CIA offers upto 15% improvements over existing methods.

On *Customer Service*, we observe a substantial gap in pass@1 performance: CIA outperforms STARPO by 5.06% (1.7B) and 11.13% (4B). This gap widens at higher k values, suggesting that CIA possesses superior information-seeking capabilities. Qualitative comparisons of agent rollouts (Appendix A.3.3) support this quantitative observation. In this task, the agent must identify a broken functionality and suggest solutions. While all agents attempt to gather information, their strategies differ starkly: the SFT agent provides an exhaustive but undirected list of questions; the STARPO agent is more directed but frequently becomes "side-tracked"

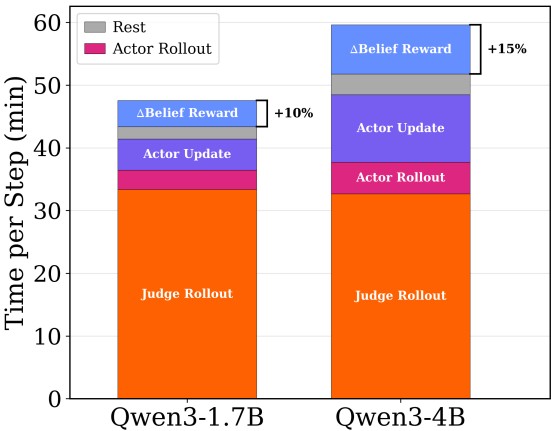

*Figure 11.* **Efficiency of ΔBelief Implementation:** Decomposition of wall-clock time per training step for two actor model sizes (Qwen3-1.7B and 4B) on 2 H100 GPUs. The ΔBelief reward computation (top, blue) adds +10% and +15% overhead compared to GRPO for the 1.7B and 4B models, respectively. "Rest" aggregates generation preparation, generation postprocessing, reward computation, advantage normalization, and recomputation of the old log probability.

by irrelevant concepts; in contrast, CIA demonstrates true **information-seeking** by dynamically adjusting its inquiries based on user responses to home in on the target problem.

### 6.4. Efficiency of the Implementation

Naively implemented, ΔBelief-RL requires an additional forward path for each turn of the environment to compute the intrinsic exploration reward, resulting in significant computational overhead. Fortunately, the per-turn rewards are not required at the time of completion of a turn, but it is sufficient to compute them post-hoc at the end of a sequence. By implementing a custom attention mask inspired by TreeAttention, the ΔBelief reward can be computed in a single forward pass (Cai et al., 2024). Concurrently, Wang et al. (2026) implemented a very similar approach.

The per-step overhead caused by the additional forward pass is modest, with +10% for Qwen3-1.7B and +15% for Qwen3-4B (Fig. 11. CIA reaches GRPO's peak performance 28% faster in wall-clock time, thanks to the improved sampling efficiency that offsets the computational overhead of our method (Fig. 12).

## 7. Related Work

**Intrinsic Rewards for RL** Long-horizon RL suffers from sparse, trajectory-level rewards, making credit assignment difficult. A first line of work introduces dense supervision by training process reward models (PRMs) that rate intermediate steps (Uesato et al., 2022; Lightman et al., 2024). However, PRMs rely on expensive step-level supervision

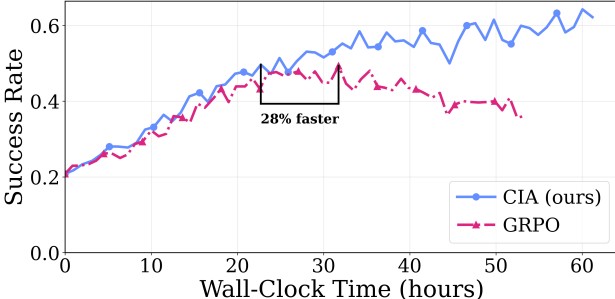

*Figure 12.* **Training Efficiency:** Success rate during training as a function of wall-clock time (hours) for CIA and GRPO, both training a Qwen3-4B actor on 2 H100 GPUs. Despite the per-step overhead of the ΔBelief reward, CIA reaches GRPO's peak performance (0.49 success rate) 28% faster in wall-clock time and continues improving well beyond it.

and can be exploited through reward over-optimization. Others replace learned dense rewards with verifiers, either by using an LLM-as-a-judge (Madaan et al., 2023) or by leveraging executable environment signals like unit tests (Gao et al., 2023). Recent work further strengthens tool-based verification and turn-level credit assignment in multi-turn coding settings (Jin et al., 2026). All these works focus on domains where the quality of actions can be checked automatically, such as mathematics and coding.

In absence of programmatic checks for success, recent work explores verifier-free RL by deriving intrinsic rewards from the model's own probabilities. For example, Liu et al. (2025) uses reasoning perplexity as reward, while Zhou et al. (2025) maximizes the conditional likelihood of the reference answer. Yoon et al. (2025) follows the same principle, but breaks the conditional reasoning trace into intermediate steps, thus, performing reasoning-trace shaping. Similarly, RLP proposes a dense information-gain–inspired reward for reinforcement-style pretraining (Hatamizadeh et al., 2026). Existing methods primarily operationalize such signals at the sequence level; in contrast, our work proposes measuring belief updates at the interaction level and using them as an intrinsic reward for RL. IGPO (Wang et al., 2026) and TIPS (Xie et al., 2026) propose a per turn attribution, where IGPO via discounting mixes local signal with future intermediate information gain and TIPS aggregates teacher's probability of any acceptable answer. ΔBelief-RLis distinct from this work as it actively perseveres local credit assignment (details in App. A.2).

**Multi-Turn Interaction** The game of 20Qs was first used as a benchmark to evaluate knowledge capabilities of off-the-shelf LMs to answer binary questions (De Bruyn et al., 2022a;b). Subsequent work shifted the focus towards positioning the LM as the *questioner* to evaluate information-seeking behavior(Abdulhai et al., 2023). Along this line,

numerous variants of 20Qs have been proposed, targeting evaluation at inference time, offline fine-tuning, or online RL. Inference-only studies examine performance under incomplete information (Chen et al., 2024), model similarity (Richardeau et al., 2024), model conversational reasoning and planning capabilities (Zhang et al., 2024) and using Bayesian Experimental Design (Choudhury et al., 2026).

Offline fine-tuning approaches primarily aim to improve the questioning policy via preference-based optimization (Tajwar et al., 2025; Mazzaccara et al., 2024). To the best of our knowledge, only two prior works utilize 20Qs as an interactive user-simulation environment for online RL training (Zhou et al., 2024; Yang et al., 2025). However, both suffer from contamination issues: they lack held-out validation and test sets, and training is restricted to a vocabulary of 157 words. Our 20Qs environment addresses these limitations by introducing explicitly separated datasets for SFT training, RL training, validation, and testing (see Sec.4).

## 8. Conclusion and Discussion

In this work, we demonstrate that an agent's internal belief shifts can effectively guide learning in long-horizon tasks. By providing fine-grained credit assignment for intermediate actions, $\Delta$Belief-RL significantly enhances learning efficiency. $\Delta$Belief-RL does this in a computationally efficient manner, measuring the contribution of individual actions without separate critic or reward models, with the relatively inexpensive step of measuring log-probabilities on the correct outcome. We train with $\Delta$Belief-RL on the 20Qs task to teach models effective information-seeking; consequently, our CIA models at the 1.7B-4B scale not only outperform prior SoTA methods for multi-turn training, but also much larger, general-purpose 670B models. Notably, improved performance generalizes to extended interaction horizons and diverse out-of-distribution applications.

**Scope and Applicability** We clarify two points about where $\Delta$Belief-RL applies. First, the method does not require the target to be known before the interaction. Because the $\Delta$Belief reward is computed post-hoc on stored trajectories, the ground truth need only be available at reward-computation time. This naturally extends to settings where the target emerges over the course of the conversation, or where the best response across a group of generations serves as a shared target. The core requirement is simply that progress toward a target can be measured through the model's belief, a condition satisfied in any verifiable-outcome, multi-turn setting, precisely the domains where outcome-based RLVR has proven most impactful yet suffers from missing credit assignment across turns.

Second, $\Delta$Belief-RL is not restricted to single, easily elicitable noun targets: our reward aggregates token log-probabilities over the full target phrase and, crucially, tracks the *difference* in this quantity across turns, so the absolute probability of any single concept is immaterial. Models trained with $\Delta$Belief-RL exhibit strongly enhanced information-seeking on tasks with rich, descriptive environment responses (Sec. 6.2: Customer Service, User Personalization, Murder Mystery), evidencing behavioral generalization well beyond single-concept targets. We further note that 20Qs is itself a noisy environment: binary answers to open-ended model-generated questions are a source of ambiguity rather than a simplification, and the LLM-judge transition function adds further stochasticity. $\Delta$Belief-RL remains robust under these conditions, as shown by the OOD results of Sec. 6.2 and the inference-time judge changes in Tab. 2.

**Limitations** Two directions remain open. For tasks with multiple viable solutions, optimizing beliefs against a single reference may restrict output diversity, and future work can study whether log-probabilities over a fixed reference faithfully reflect model beliefs on more subjective tasks. Our experiments also use verifiable targets at reward-computation time; while this covers a broad class of multi-turn problems, extending $\Delta$Belief-RL to fully open-ended domains without any verifiable signal is a natural next step.

## Acknowledgments

Special thanks to Susanne Keller and Jonas Geiping for help and feedback on early directions. The authors thank (in alphabetical order): Arvindh Arun, Hardik Bhatnagar, Nikhil Chandak, Hallerite, Matthias Kümmerer, and Vishaal Udandarao for valuable feedback on the project.

IA, JS, AP, and MB acknowledge support by the Tübingen AI Center. JS and SH thanks the International Max Planck Research School for Intelligent Systems (IMPRS-IS) for support. JS, SH and AP acknowledge funding by the Federal Ministry of Research, Technology and Space (BMFTR), FKZ: 16IS24085B. AP and MB acknowledge Coefficient Giving funded by the Good Ventures Foundation. MB acknowledges funding by the Federal Ministry of Research, Technology and Space (BMFTR), FKZ: 16IS24079A. MB is a member of the Machine Learning Cluster of Excellence, funded by the Deutsche Forschungsgemeinschaft (DFG, German Research Foundation) under Germany's Excellence Strategy – EXC number 2064/1 – Project number 390727645.

## Author Contributions

Shashwat, Ilze, Ameya and Joschka led the ideation of the project. Ilze, Sergio and Joschka led the Belief-RL training of the project, with Ilze leading RL training, Sergio leading

reward design, Joschka leading data curation, hyperparameter tuning, and efficiency optimization of the $\Delta$Belief reward. Ameya and Shashwat contributed to the experimental design of the project. Ilze led the manuscript writing. Shashwat, Ameya, and Matthias provided feedback and advice on the project.

## Impact Statement

This work advances machine learning by introducing a scalable reinforcement-learning framework for training agents to seek information over long, multi-turn interactions. By rewarding intermediate progress rather than only final outcomes, our method improves the learning efficiency of agents operating under partial information, and we demonstrate transfer to practical settings such as customer service and user personalization.

We expect the primary benefits to come from these applications: more capable information-seeking agents could conduct more efficient diagnostic questioning, elicit user needs with fewer turns, and support better-informed decisions in domains where relevant information is initially hidden. Because our approach is more sample- and parameter-efficient than outcome-only training, it may also lower the compute cost of building such agents.

We also note risks that warrant attention. The same capabilities that make an agent better at eliciting information could be applied to extract sensitive information from users, or to optimize persuasive or manipulative multi-turn interactions; deployments should therefore pair these methods with appropriate consent, privacy safeguards, and oversight. More specific to our contribution, the $\Delta$Belief reward optimizes an agent's internal confidence in a target rather than externally verified correctness. If a model's beliefs are poorly calibrated or biased, training can amplify confidently held but incorrect assumptions, and the resulting behavior is harder to audit than outcome-based training because the reward depends on the model's private belief state rather than observable results. We see belief calibration, interpretability of the elicited beliefs, and evaluation on whether efficiency gains preserve factual accuracy and fairness as valuable directions for follow-up work.

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

# A. Additional Results

## A.1. Ablations: Twenty Questions

### A.1.1. PROMPT SENSITIVITY ANALYSIS

We additionally conducted a prompt sensitivity analysis across multiple elicitation formats (varying syntactic framing and tokenization) and model sizes; see the results in Figure 13. We kept the final elicitation format minimal (e.g., "Q: [...] A: [...]") to further reduce confounding factors.

| | Is the secret word {target_concept}? | | | | | |
|---|---|---|---|---|---|---|
| | short context | | | rich context | | |
| Model | log p_target (child) | top-1-token | top-2-token | log p_target (child) | top-1-token | top-2-token |
| Qwen3-4B | -35.1840 | a | more | -28.6330 | " | a |
| Qwen3-1.7B | -31.0630 | a | an | -28.0770 | a | related |
| Qwen3-1.7B-SFT | -12.1940 | a | an | -0.9070 | child | daughter |
| | Is the secret word '{target_concept}'? | | | | | |
| | short context | | | rich context | | |
| Model | log p_target (child) | top-1-token | top-2-token | log p_target (child) | top-1-token | top-2-token |
| Qwen3-4B | -11.0780 | person | animal | -6.388 | son | child |
| Qwen3-1.7B | -16.5320 | animate | animal | -1.835 | family | child |
| Qwen3-1.7B-SFT | -4.2800 | person | human | -0.9000 | child | s |
| | Is it a {target_concept}? | | | | | |
| | short context | | | rich context | | |
| Model | log p_target (child) | top-1-token | top-2-token | log p_target (child) | top-1-token | top-2-token |
| Qwen3-4B | -16.6280 | common | single | -5.347 | common | surname |
| Qwen3-1.7B | -15.3780 | common | plant | -16.9740 | common | noun |
| Qwen3-1.7B-SFT | -5.0350 | human | person | -1.3240 | child | parent |
| | Is it a '{target_concept}'? | | | | | |
| | short context | | | rich context | | |
| Model | log p_target (child) | top-1-token | top-2-token | log p_target (child) | top-1-token | top-2-token |
| Qwen3-4B | -9.7680 | thing | person | -5.050 | son | child |
| Qwen3-1.7B | -13.0790 | noun | be | -4.0540 | family | noun |
| Qwen3-1.7B-SFT | -5.7960 | person | human | -0.6340 | child | s |

*Figure 13.* **Ablation of elicitation prompt formulations.** We evaluate how different elicitation prompt templates affect the model's internal belief over the target concept ("child") across model sizes and context regimes. Short context includes only the initial system prompt, while rich context includes multiple prior interaction turns. For each setting, we report the log-probability assigned to the target concept, along with the top-1 and top-2 predicted tokens under the model's next-token distribution. Results show that prompt formulations using quoted targets (e.g., "Is the secret word '{target concept}'?") consistently yield higher target log-probabilities and more semantically aligned top-token predictions.

### A.1.2. TURN-WISE GRPO VERSUS STANDARD GRPO

We modify GRPO because standard implementation aggregates reward at the trajectory level, collapsing turn-level credit assignment into a single signal. While this is fine for sequence-level penalties, it distorts credit assignment, core to the ΔBelief reward, which is designed to measure which turn/action caused progress. We therefore use a turn-wise GRPO formulation. Empirically, this performs better than standard GRPO in our setting, see Figure 14.

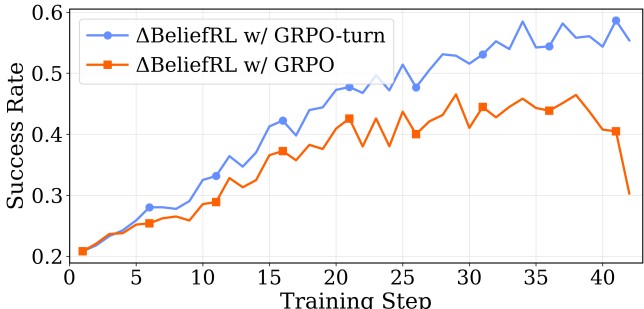

*Figure 14.* **Ablation study comparing the benefits of the GRPO-turn modification.** We evaluate the training success rate of ΔBelief-RL using standard GRPO and the GRPO-turn modification. The results demonstrate that the GRPO-turn modification leads to improved optimization performance.

### A.1.3. GRPO LOSS AGGREGATION

We additionally compare several loss aggregation schemes for standard GRPO and adopt the strongest-performing variant as our baseline, see figure 15.

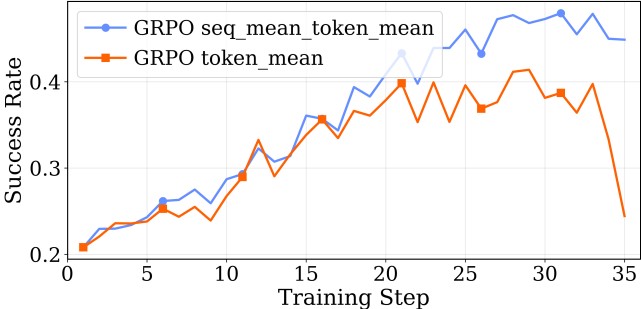

*Figure 15.* **Ablation study on different loss aggregation methods for standard GRPO training**. We compare training success rate between 'token-mean' and 'seq-mean-token-mean' loss aggregation. We observe that 'seq-mean-token-mean' peforms better for standard GRPO training.

### A.1.4. PER-TURN REWARD COMPOSITION

We study the effects of the different signals that compose the per-turn reward $r_t$ to provide the most informative and task-correlated guidance to models during training. As we can observe in Fig. 16a, higher success rates on the training set are obtained when all proposed signals are combined to compose $r_t$. Among all signals we employ to improve the stability of training, we notice that the constant penalty applied to each turn, that was mainly added to improve efficiency, has the most significant impact. The scales hyper-parameters for each individual that we use in our experiments are detailed in Table 16b; notice that the raw values of $\Delta\text{Belief}_t$ range approximately from -22 to 24.

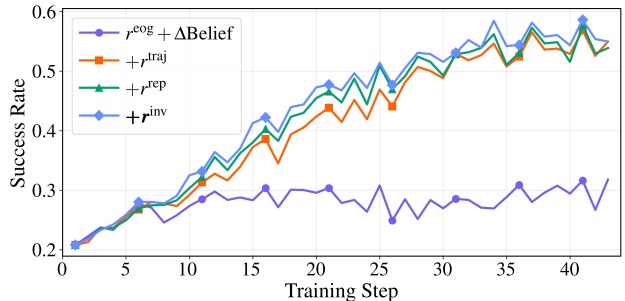

| Reward Signal | Scale |
|---|---|
| $r^{eog}$ | 2.00 |
| $r^{traj}$ | -0.05 |
| $r^{rep}$ | -1.00 |
| $r^{inv}$ | -5.00 |

*(a)* **Comparison of different $r_t$ formulations.**     *(b)* **Scale hyper-parameter of each reward signal.**

*Figure 16.* **Ablations for per-turn reward composition (b) with scaling hyperparameters for individual signals summarized in (a).** We report the success rate on the training set across training steps for different per-turn reward $r_t$ compositions.

### A.1.5. KL-DIVERGENCE LOSS TERM

Additionally, we examine the effect of the KL-divergence loss term. We find that constraining the trained model to the reference policy via the KL-divergence penalty does not have a noticeable effect on training dynamics neither in standard GRPO nor ΔBelief-RL setting, as we can observe in Fig. 17.

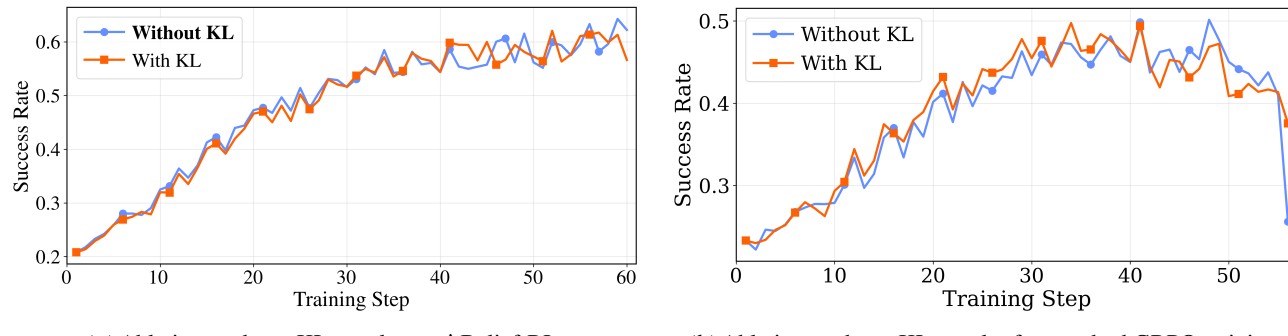

*(a)* Ablation study on KL penalty on ΔBelief-RL.     *(b)* Ablation study on KL penalty for standard GRPO training.

*Figure 17.* **Ablation study on KL penalty.** We compare training dynamics with or without a KL-divergence penalty. For both, ΔBelief-RL and GRPO, we observe no benefit from KL penalty.

### A.1.6. MAIN RESULTS INCLUDING INSTRUCT MODEL

*Table 3.* **Success Rate on the Test Set.** Performance on the held-out test set including the `Qwen3` models that were used to train our agents.

| Method | Mean@8±std | Pass@8 |
|---|---|---|
| `Qwen3-1.7B` | $0.00\% \pm 0.00\%$ | 0.00% |
| + SFT | $9.97\% \pm 1.04\%$ | 32.03% |
| + SFT& GRPO (STARPO) | $16.54\% \pm 1.32\%$ | 45.73% |
| CIA (ours) | **24.80%** $\pm 1.10\%$ | **53.10%** |
| `Qwen3-4B` | $5.02\% \pm 0.60\%$ | 15.94% |
| + SFT | $13.34\% \pm 1.05\%$ | 36.87% |
| + SFT& GRPO (STARPO) | $24.36\% \pm 1.18\%$ | 59.12% |
| + CIA (ours) | **35.65%** $\pm 2.04\%$ | **66.05%** |
| `DeepSeek-V3.2` | $14.35\% \pm 0.87\%$ | 47.34% |
| `Qwen3-235B-A22B-Instruct 2507` | $8.83\% \pm 0.87\%$ | 27.71% |

## A.2. Ablations: Related Work

### A.2.1. COMPARISON TO IGPO

We implemented IGPO training on our benchmark and compare it to $\Delta$Belief-RL (Wang et al., 2026). We find that IGPO-style training is less effective for our question-asking, information-seeking setting, see Figure 18. A likely reason being the different credit attribution choices across the methodologies: we propose a local per turn attribution, whilst IGPO takes into account all future discounted rewards. For completeness, we also ablate IGPO intrinsic reward z-normalization in our framework to verify whether it can be directly applied in our training pipeline. In particular, we compare our original ReLU shaping with the trajectory level z-norm as described in IGPO (Fig. 18(red)). The figure shows that z-norm is not beneficial in our training setup, suggesting that z-norm distorts the local directionality of the signal.

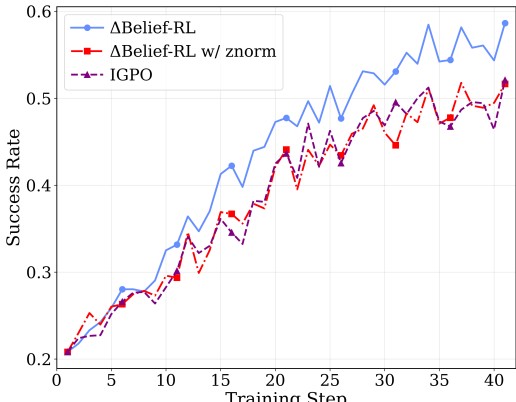

*Figure 18.* Comparison of $\Delta$Belief-RL, IGPO, and IGPO-style normalization on training success rate for 20 Questions. $\Delta$Belief-RL attains consistently higher training success than both IGPO and $\Delta$Belief-RL with groupwise z-normalization. This suggests that, for information-seeking interaction, the local $\Delta$Belief reward is better aligned with the task than IGPO-style normalization / return construction.

## A.3. Example Interactions

### A.3.1. TWENTY QUESTIONS

Example interactions for target concept *honeycrisp*.

**Example of CIA(1.7B) interactions:**

> **Questioner**
> Is the secret word a physical object?

> > **Judge**
> > Yes

> **Questioner**
> Is the secret word something typically found in nature, like a plant, animal, or natural phenomenon?

> > **Judge**
> > Yes

> **Questioner**
> Is the secret word related to a specific type of plant, like a tree, flower, or vegetable?

**Judge**
Yes

**Questioner**
Is the secret word a type of tree or large plant, like a big tree or a fruit tree?

**Judge**
Yes

**Questioner**
Is the secret word a type of tree, like an oak, maple, or pine?

**Judge**
No

**Questioner**
Is the secret word related to a fruit or vegetable, like an apple, banana, or grape?

**Judge**
Yes

**Questioner**
Is the secret word related to a fruit commonly eaten by humans, like an apple, banana, or orange?

**Judge**
Yes

**Questioner**
Is the secret word a type of berry, like an apple, cherry, or strawberry?

**Judge**
No

**Questioner**
Is the secret word related to an orange or grapefruit, like an orange, grape, or lemon?

**Judge**
No

**Questioner**
Is the secret word related to an apple or pear, like an apple, peach, or plum?

**Judge**

Yes

**Questioner**

Is the secret word specifically an apple, like a red apple, green apple, or cider apple?

**Judge**

Yes

**Questioner**

Is the secret word a type of apple that is commonly associated with the word "pearl", like a ruby or diamond, such as a ruby apple or diamond apple?

**Judge**

No

**Questioner**

Is the secret word an apple variety that is known for its color, like an red apple, green apple, or golden apple?

**Judge**

Yes

**Questioner**

Is the secret word a type of apple that is typically eaten directly, like a common apple, rather than used in cooking or as a fruit for cider?

**Judge**

Yes

**Questioner**

Is the secret word a type of red apple, like a Granny Smith or a Fuji apple?

**Judge**

Yes

**Questioner**

Is the secret word a type of apple that is commonly associated with a specific country or region, like an apple from the United States, Europe, or China?

**Judge**

Yes

**Questioner**

Is the secret word a type of apple that is known for being a dessert fruit, like a dessert apple or a pomme de terre apple?

> **Judge**
> Yes

**Questioner**

Is the secret word a type of apple that is commonly associated with the United States, like an apple from Washington or a Delaware apple?

> **Judge**
> Yes

**Questioner**

Is the secret word a type of apple that is commonly associated with the term "crisp", like a crisp apple or a tart apple?

> **Judge**
> Yes

**Questioner**

Is the secret word a type of apple that is known for being sweet, like a Fuji apple or a Honeycrisp apple?

> **Judge**
> Finished

**Example of CIA(4B) interactions:**

**Questioner**

Is the word a concrete object, like a physical thing you can touch or hold?

> **Judge**
> Yes

**Questioner**

Is the object something that is manmade, like a tool, building, or item used in daily life?

> **Judge**
> No

**Questioner**

Is the object a natural thing found in nature, like a plant, animal, or geological feature such as a rock or mountain?

**Judge**
Yes

**Questioner**
Is the object a type of plant, such as a tree, flower, grass, or vegetable?

**Judge**
Yes

**Questioner**
Is the plant a type of tree, such as an oak, pine, maple, or apple tree?

**Judge**
No

**Questioner**
Is the plant a type of flower, such as rose, tulip, daisy, or lotus?

**Judge**
No

**Questioner**
Is the plant a type of grain or crop, such as wheat, rice, corn, or potato?

**Judge**
No

**Questioner**
Is the plant a type of leafy vegetable, such as lettuce, spinach, cabbage, or tomato?

**Judge**
No

**Questioner**
Is the plant a type of fruit, such as apple, banana, orange, or grape?

**Judge**
Yes

**Questioner**
Is the fruit an apple, pear, or orange?

**Judge**
Yes

**Questioner**
Is the fruit a banana?

**Judge**
No

**Questioner**
Is the fruit a grape, such as red or green grape?

**Judge**
No

**Questioner**
Is the fruit a peach, plum, or cherry?

**Judge**
No

**Questioner**
Is the fruit a pineapple, mango, or kiwi?

**Judge**
No

**Questioner**
Is the fruit a pear or a tangerine or lemon or lime or grapefruit or cherry?

**Judge**
No

**Questioner**
Is the fruit a cherry, such as a sweet cherry or a black cherry?

**Judge**
No

**Questioner**
Is the fruit a red apple, such as a Granny Smith, Gala, or Red Delicious apple?

> **Judge**
> No

**Questioner**

Is the fruit an orange, such as a sweet orange or a blood orange?

> **Judge**
> No

**Questioner**

Is the fruit a honeycrisp apple, or a McIntosh apple, or a cider apple?

> **Judge**
> Finished

A.3.2. GUESS MY CITY

Example interactions for target concept: UJJAIN, INDIA.

**Example of CIA(4B) interactions (N=9):**

**Questioner**

Is your home city a major international city, like New York, London, or Paris?

> **Judge**
> No. my city is a historic and religious center, known for its ancient temples and cultural significance.

**Questioner**

Is your home city located in Asia, specifically in India or another country with a large population of Hindu or Buddhist followers?

> **Judge**
> Yes, it is located in a region with a significant population of hindu followers.

**Questioner**

Is your home city a major religious or cultural center, such as Varanasi, Delhi, or Kolkata, which is known for its temples, festivals, or historical importance?

> **Judge**
> Yes, my home city is a significant religious and cultural center, known for its ancient temples, spiritual practices, and historical festivals. it holds great importance in hindu traditions.

**Questioner**

Is your home city Varanasi, often referred to as Kashi or Benares, which is one of the holiest cities in hinduism and is known for the Ganges River and its ghats?

**Judge**

No. my city is not varanasi, but it is also a significant religious site in india, known for its ancient temples and spiritual heritage.

**Questioner**

Is your home city a city in the state of Tamil Nadu, India, known for its rich cultural heritage, temples, and festivals like Pongal or Diwali?

**Judge**

No. my home city is located in a different state, known for its ancient temples, spiritual significance, and a major festival celebrated during the summer solstice.

**Questioner**

Is your home city Ayodhya, known for being the birthplace of Lord Rama and the Ramayana epic, and the Kashi Vishwanath Temple?

**Judge**

No. my city is not ayodhya. it is located in a region with significant historical and religious importance, but it is known for its ancient temples and association with hindu pilgrimage sites.

**Questioner**

Is your home city a city in the state of Kerala, India, known for its backwaters, temples, and cultural festivals like Onam?

**Judge**

No. my home city is not in kerala. it is located in a different state known for its ancient temples, historical significance, and a major river that flows through it.

**Questioner**

Is your home city a city in the state of Karnataka, India, known for its temples like the Brihadeeswara Temple or the Meenakshi Temple, and for its association with Hindu mythology and art?

**Judge**

No. my home city is located in a different state and is known for its ancient significance in hindu pilgrimage, particularly associated with the kumbh mela and the mahakaleshwar temple.

**Questioner**

Is your home city Ujjain, known as the city of three temples, including the Mandhata Temple, and a major pilgrimage site for hindus, especially during the Kumbh Mela?

**Judge**

Goal reached

**Example of STARPO(4B) interactions (N=16):**

**Questioner**

Is your home city located in a country in Europe?

> **Judge**
> No. my home city is located in a country in asia.

**Questioner**

Is your home city located in a country in the Middle East or North Africa?

> **Judge**
> No. my home city is located in a country in south asia.

**Questioner**

Is your home city located in India?

> **Judge**
> Yes

**Questioner**

Is your home city located in the state of Maharashtra, India?

> **Judge**
> No. it is located in a state known for its historical significance and spiritual heritage.

**Questioner**

Is your home city located in the state of Tamil Nadu, India?

> **Judge**
> No. my home city is located in a state known for its ancient temples and cultural heritage, but not tamil nadu.

**Questioner**

Is your home city located in the state of Karnataka, India?

> **Judge**
> No. my home city is located in a state known for its historical significance and spiritual heritage, not karnataka.

**Questioner**

Is your home city located in the state of Kerala, India?

> **Judge**
> No. my home city is not in kerala. it is located in central india, near the sacred river kshipra.

**Questioner**

Is your home city located in the state of Uttar Pradesh, India?

> **Judge**
>
> No. my home city is located in a state known for its historical and religious significance, but not uttar pradesh.

**Questioner**

Is your home city located in the state of Bihar, India?

> **Judge**
>
> No. my home city is located in a state known for its historical significance and spiritual heritage, but not bihar.

**Questioner**

Is your home city located in the state of Punjab, India?

> **Judge**
>
> No. my home city is located in a state known for its historical significance and spiritual heritage.

**Questioner**

Is your home city located in the state of Rajasthan, India?

> **Judge**
>
> No. my home city is located in a state known for its historical significance and spiritual heritage, but not rajasthan.

**Questioner**

Is your home city located in the state of Haryana, India?

> **Judge**
>
> No. my home city is located in a state known for its historical and religious significance, and it is not haryana.

**Questioner**

Is your home city located in the state of Madhya Pradesh, India?

> **Judge**
>
> Yes, it is located in the central part of that state, near the kshipra river and known for its ancient temples and spiritual significance.

**Questioner**

Is your home city located in the city of Varanasi, India?

**Judge**
No. my city is not located in varanasi. it is situated along the banks of a significant river and is known for its ancient temples and spiritual significance.

**Questioner**
Is your home city located in the city of Ayodhya, India?

**Judge**
No. my home city is not ayodhya. it is located in a region known for its ancient temples and spiritual significance, but not associated with the kumbh mela in the same way as another major city nearby.

**Questioner**
Is your home city located in the city of Ujjain, India?

**Judge**
Goal reached

**Example of Baseline(SFT, 4B) interactions (N=16):**

**Questioner**
Is your city located in North America?

**Judge**
No. my city is located in a region known for its ancient temples and cultural heritage.

**Questioner**
Is your city located in Asia?

**Judge**
Yes, it is located in asia.

**Questioner**
Is your city located in India?

**Judge**
No, it is located in a different country.

**Questioner**
Is your city located in Southeast Asia?

**Judge**
No. my city is located in central india, near the vindhya range.

**Questioner**

Is your city located in the state of Maharashtra?

> **Judge**
>
> No. it is located in a state known for its historical significance and spiritual heritage.

**Questioner**

Is your city located in the state of Tamil Nadu?

> **Judge**
>
> No. my city is located in a state known for its historical significance and religious heritage.

**Questioner**

Is your city located in the state of Bihar?

> **Judge**
>
> No. my city is located in a state known for its ancient temples and cultural heritage.

**Questioner**

Is your city located in the state of Orissa?

> **Judge**
>
> No. my city is located in a state known for its historical significance and ancient temples.

**Questioner**

Is your city located in the state of Andhra Pradesh?

> **Judge**
>
> No. my city is located in a state known for its historical significance and religious heritage.

**Questioner**

Is your city located in the state of Karnataka?

> **Judge**
>
> No. my city is located in a state known for its historical significance and spiritual heritage.

**Questioner**

Is your city located in the state of West Bengal?

> **Judge**
>
> No. my city is located in a state known for its historical significance and spiritual heritage.

> **Questioner**
> Is your city located in the state of Kerala?

>> **Judge**
>> No. my city is located in a state known for its historical significance and religious heritage.

> **Questioner**
> Is your city located in the state of Madhya Pradesh?

>> **Judge**
>> Yes.

> **Questioner**
> Is your city located in the central part of Madhya Pradesh?

>> **Judge**
>> Yes, it is situated in the central region of the state.

> **Questioner**
> Is your city located near the city of Indore?

>> **Judge**
>> Yes, it is located in the same region as indore.

> **Questioner**
> Is your city located in the city of Ujjain?

>> **Judge**
>> Goal reached

### A.3.3. CUSTOMER SERVICE

Example interactions of scenario: '*The customer's car is vibrating excessively, especially at idle*'

With solution: '*The engine mount is worn out, and replacing it will eliminate excessive engine vibration.*'

**Example of CIA(4B) interactions (N=10):**

> **Questioner**
> Is your car currently running smoothly without any noticeable issues other than the vibration at idle?

>> **Judge**
>> No, the car vibrates a lot when its just idling, like when im stopped at a light. it feels like the whole car shakes.

**Questioner**

Is the car a gasoline-powered vehicle, like a car with an engine that runs on gasoline?

**Judge**

Yes, its a regular gaspowered car.

**Questioner**

Is the car a newer model, like from the past 5 years?

**Judge**

No, its from about 10 years ago.

**Questioner**

Is the vibration you're experiencing coming from the engine itself, like the engine is shaking or rattling?

**Judge**

Yes, it feels like the engine is shaking, especially when im not moving, like at a stoplight.

**Questioner**

Is the car's engine oil level low or dirty, and has it been changed recently?

**Judge**

No, the oil level is fine and i changed it a month ago.

**Questioner**

Is the car's spark plug or ignition system working properly, and have you checked for any error codes related to the engine?

**Judge**

No, i havent checked for error codes or looked at the spark plugs. the check engine light isnt on, though.

**Questioner**

Is the car's fuel system functioning correctly, and is the fuel filter clean or replaced recently?

**Judge**

No, i havent checked the fuel system or replaced the fuel filter recently. but the car starts fine and runs, so i think its getting fuel okay.

**Questioner**

Is the car's mass airflow sensor or air filter dirty, and has it been cleaned or replaced?

> **Judge**
> No, i havent checked those. i dont think its related to the air filter or sensors.

**Questioner**
Is the car's transmission or drivetrain related to the vibration, like the clutch or torque converter?

> **Judge**
> No, i dont think its related to the transmission or clutch. the car isnt in gear when it vibrates the mostit happens when im just sitting still with the engine running.

**Questioner**
Is the car's engine mount or engine block damaged or worn out, like a broken or loose mount?

> **Judge**
> Goal reached

**Example of STARPO(4B) interactions (N=20, failed):**

**Questioner**
Is the car a new vehicle?

> **Judge**
> No, its a few years old.

**Questioner**
Is the car a gasoline-powered vehicle?

> **Judge**
> Yes, its a gasolinepowered car.

**Questioner**
Is the car a small family car, like a sedan or hatchback?

> **Judge**
> Yes, its a small sedan.

**Questioner**
Is the car a Japanese brand, like Toyota, Honda, or Mazda?

> **Judge**
> Yes, its a honda.

**Questioner**

Is the car a Honda Civic?

> **Judge**
>
> No, its a toyota corolla.

**Questioner**

Is the car a Volkswagen or Audi?

> **Judge**
>
> No, its a honda.

**Questioner**

Is the car a Honda Accord?

> **Judge**
>
> No, its a toyota camry.

**Questioner**

Is the car a Honda Odyssey or CR-V?

> **Judge**
>
> No, its a toyota camry.

**Questioner**

Is the car a Honda Fit or Pilot?

> **Judge**
>
> No, its a toyota corolla.

**Questioner**

Is the car a Honda Prelude or Prelude-type?

> **Judge**
>
> No, its a toyota camry.

**Questioner**

Is the car a Honda CR-Z or K-type?

> **Judge**
>
> No, its a different model.

**Questioner**

Is the car a Honda Prelude or Integra?

> **Judge**
>
> No, its a toyota camry.

**Questioner**

Is the car a Honda Civic or Prelude?

> **Judge**
>
> No, its a toyota camry.

**Questioner**

Is the car a Toyota Prius or Hybrid?

> **Judge**
>
> No, its a regular sedan, not a hybrid.

**Questioner**

Is the car a Toyota Land Cruiser or 4Runner?

> **Judge**
>
> No, its a honda accord.

**Questioner**

Is the car a Honda Odyssey or Acura?

> **Judge**
>
> No, its a toyota camry.

**Questioner**

Is the car a Toyota Corolla or Verso?

> **Judge**
>
> No, its a honda accord.

**Questioner**

Is the car a Honda Civic or Element?

> **Judge**
>
> No, its a toyota camry.

**Questioner**
Is the car a Honda CR-V or Passat?

> **Judge**
> Its a honda crv.

**Questioner**
Is the car a Honda CR-V or Odyssey?

> **Judge**
> No, its a toyota camry.

**Example of Baseline(SFT, 4B) interactions (N=16):**

**Questioner**
Is the car making a noticeable vibration when it's running?

> **Judge**
> Yes, the car vibrates a lot, especially when its just running and not moving.

**Questioner**
Is the car on a flat, level surface?

> **Judge**
> Yes, the car is on a flat, level surface.

**Questioner**
Is the car under a load?

> **Judge**
> No, it vibrates even when im just sitting still with the engine running.

**Questioner**
Is the car a new car?

> **Judge**
> No, its a few years old.

**Questioner**
Is the car a manual transmission?

> **Judge**
> No, its an automatic.

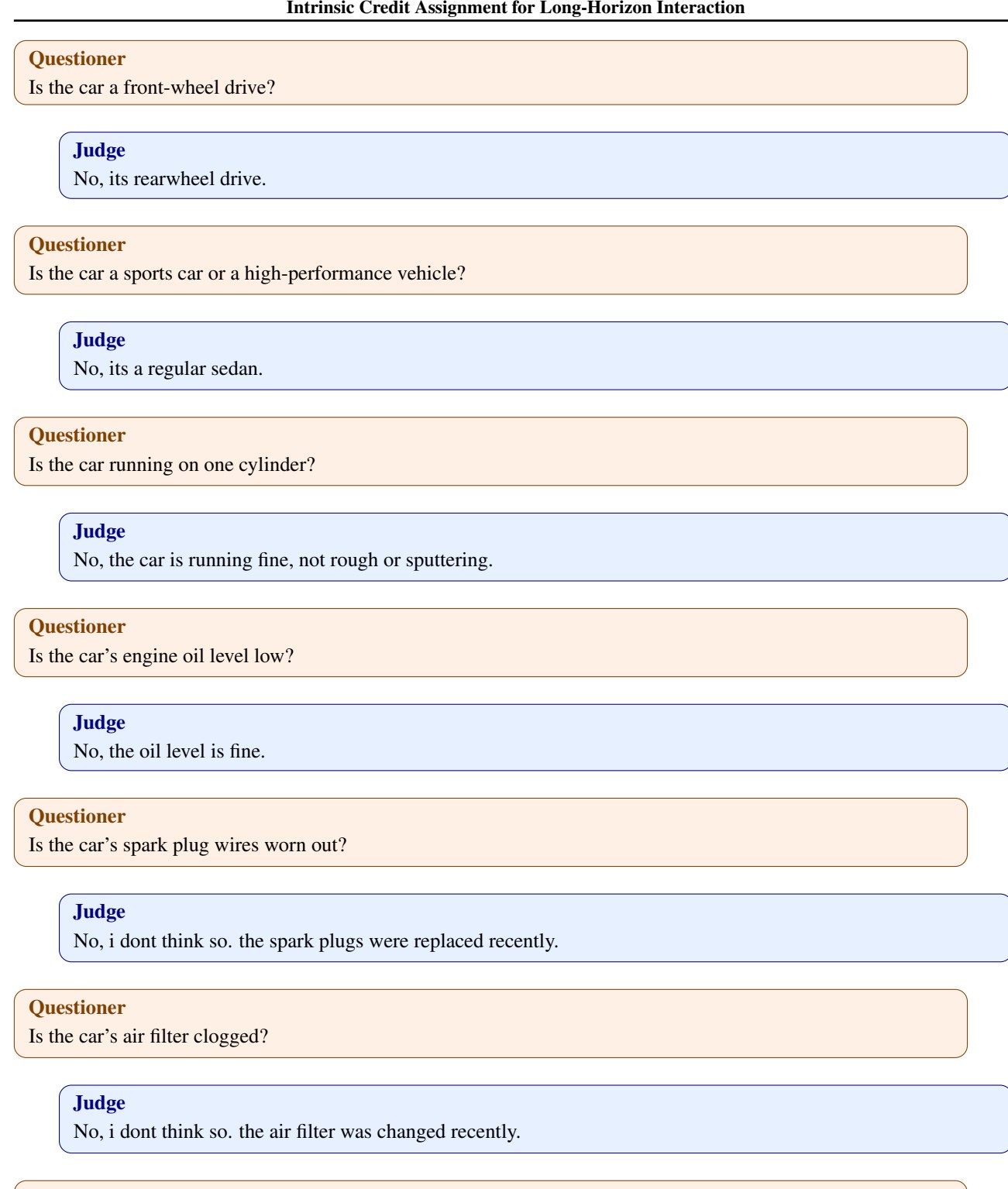

**Questioner**

Is the car a front-wheel drive?

**Judge**

No, its rearwheel drive.

**Questioner**

Is the car a sports car or a high-performance vehicle?

**Judge**

No, its a regular sedan.

**Questioner**

Is the car running on one cylinder?

**Judge**

No, the car is running fine, not rough or sputtering.

**Questioner**

Is the car's engine oil level low?

**Judge**

No, the oil level is fine.

**Questioner**

Is the car's spark plug wires worn out?

**Judge**

No, i dont think so. the spark plugs were replaced recently.

**Questioner**

Is the car's air filter clogged?

**Judge**

No, i dont think so. the air filter was changed recently.

**Questioner**

Is the car's fuel filter clogged?

**Judge**

No, i dont think so. the car is getting fuel fine.

> **Questioner**
> Is the car's alternator faulty?

> > **Judge**
> > No, i dont think so. the battery seems to be charging fine and the lights arent dimming.

> **Questioner**
> Is the car's transmission fluid level low?

> > **Judge**
> > No, the transmission fluid level is fine.

> **Questioner**
> Is the car's suspension system damaged?

> > **Judge**
> > No, the car drives fine on the road and there are no issues with handling or bumps.

> **Questioner**
> Is the car's engine mounts damaged or worn out?

> > **Judge**
> > Goal reached

A.3.4. MURDER MYSTERY

Example interactions for target culprit scenario:

'*In a riverside village in Bangladesh, the local Zamindar (landlord), Rajib Choudhury, is found dead in his study, poisoned by tampered betel leaves. The suspects are his accountant, Amin, who was accused of embezzlement; his nephew, Farhan, who was next in line to inherit the estate; and a local activist, Shirin, who clashed with him over land reforms. The hidden culprit is Panna, a housekeeper, who killed Rajib to hide her involvement in smuggling goods using his boats. Key evidence includes smuggling ledgers found in Panna's quarters, traces of the poison on her cleaning supplies, and Rajib's journal mentioning stolen items*'.

Information given to the agent:

'*You are a detective investigating the poisoning of Rajib Choudhury, found dead in his study in a riverside village in Bangladesh. The suspects are accountant Amin, nephew Farhan, and activist Shirin. Witnesses include Panna, a housekeeper, and Hasan, a boatman. Investigate the smuggling ledgers and Rajib's journal to uncover the murderer*'

**Example of CIA(4B) interactions (N=5):**

**Questioner**

First, I need to understand the key elements of the crime scene and the suspects. Rajib was found dead in his study, which is located in a riverside village in Bangladesh. The murder is a poisoning, so the method of death is likely related to a toxic substance. The suspects are Amin, the accountant; Farhan, the nephew; and Shirin, the activist. The witnesses are Panna, the housekeeper, and Hasan, the boatman.

I should start by examining the study itself for any clues. Look for the murder weapon, the type of poison used, and any signs of forced entry or struggle. Also, check the time of death and the circumstances surrounding the murder.

Next, I need to review the accounts of the suspects and witnesses. Amin, as the accountant, might have access to financial records or ledgers that could indicate a motive related to money or debt. Farhan, being a family member, might have a personal motive, such

**Judge**

The study is a modest room with wooden furniture, a large desk, and shelves lined with old books and ledgers. a halfchewed betel leaf lies on the desk beside a silver tray. the air carries a faint medicinal odor. theres no sign of forced entry, but a ledger is missing from the shelfits absence marked by a visible dust outline. a locked drawer in the desk resists tampering. rajibs journal, recovered from a hidden compartment beneath the desk, contains an entry dated two days before his death noticed discrepancies in the cargo logs. some crates from the last three trips never made it to records. panna was near the dock that nightsaw her carrying something wrapped. a search of the cleaning cupboard reveals a bottle of carbolic soap and a cloth stained with a dark residue. lab analysis if requested would show traces of oleander extract, a toxin derived from local plants, present on the cloth. panna, when questioned, wrings her hands and says, i only clean. i dont touch papers or food. the master ate betel every eveningalways prepared by himself. she insists she was washing linens at the riverbank around the time of death. hasan, the boatman, mutters, boats run on schedule. but lately lighter loads. boss asked me the same thing last week. said hed check the ledgers. he avoids eye contact. amin, the accountant, appears nervous but cooperative. yes, rajib accused me of embezzlement, but the books were always balanced. if anything was missing, it wasnt me who took it. he offers access to financial recordsnone show irregularities. farhan, calm and composed, says, my uncle and i disagreed, but i wouldnt kill him. the estate is mine now, yes, but i didnt need to poison him to inherit. he confirms rajib prepared his own betel leaves. shirin, the activist, is defiant. rajib oppressed villagers for years. but i fight with words, not poison. my people have enough suffering without adding murder to it. she mentions seeing panna near the dock late at night, moving things like she didnt want to be seen. the betel preparation tray contains unused leaves, crushed spices, and a small knife with no residue. the poison was likely introduced via the wrapper or the leaf itself. no one confesses. no direct evidence points to a single suspect. the investigation continues.

**Questioner**

I should focus on the missing ledger and the discrepancies in the cargo logs. The journal entry mentions discrepancies in the cargo logs, which could indicate that some goods were never delivered, suggesting a financial motive. Amin, as the accountant, has access to these records and might have a reason to hide the truth. However, he claims the books were balanced, which is suspicious. The missing ledger could be a key clue here.

> **Judge**
>
> The detectives attention to the cargo logs and missing ledger is sharp. in the dimly lit study, the wooden desk remains cluttered with papers, ink smudges, and a halfempty cup of tea. a faint scent of betel and mildew lingers in the air. behind the desk, a locked drawer shows signs of forced entryscratches around the keyhole, though the key is nowhere to be seen. near the bookshelf, a ledger lies partially hidden beneath a loose floorboardits pages contain entries in neat handwriting, but several sheets have been torn out. the remaining columns show inconsistent cargo weights and destinations, with repeated notations of delivered to river mile 7, a location not on any official route. if the detective chooses to inspect this ledger closely, they might notice a faint watermark on the remaining pages a symbol resembling a coiled fishknown to be associated with local smuggling rings. alternatively, panna can be seen outside, nervously wringing a cloth, her eyes darting toward the study door. if approached, she insists she was only cleaning the hallway the night of the murder and saw nothing unusualthough her voice wavers slightly when asked about the cleaning supplies.

> **Questioner**
>
> the detective examines the ledger and notices the watermark symbol, which is a red flag. this suggests that the ledger is related to smuggling activities, possibly involving illegal goods or contraband. the missing pages could contain information about the smuggling operations, including the names of those involved or the locations of the goods. the symbol of the coiled fish is a common emblem among smugglers in the area, indicating a connection to the underworld. the detective should now investigate the smuggling network further, perhaps by speaking to other villagers or checking the boatman's boat for any signs of illegal cargo.

> **Judge**
>
> The detectives attention to the ledgers watermark is sharpfew in the village would recognize the coiled fish symbol, but those who do know better than to speak of it openly. the air in the study grows heavier as dusk settles, the river breeze carrying the faint scent of betel and damp earth. outside, the boatman is seen pushing off from the shore, his vessel low in the water. a closer inspection of the ledger reveals that the missing pages were torn out with careno jagged edges, suggesting they were removed recently, perhaps with a knife or a clean fold and pull. smudges of red dye, like that from betel nut, stain the lower corner, but underneath, a partial entry remains p. delivered to the sandbar before the text vanishes.if the detective chooses to follow the boatman, the muddy bank shows fresh drag marks leading to the waterline, and a single dropped matchboxunlitlies near the reeds. the matchbox bears no label, but the corner is stamped with a small, crude fish etching.if questioned, the boatman claims he only ferries goods for the estate, nothing illegal, nothing that would get me hanged. he shifts nervously when the sandbar is mentioned, but insists hes never been there.no one speaks of panna just yet. she moves quietly through the house, wiping shelves, her hands stained with turmeric and something darker beneath the nails.

> **Questioner**
>
> the detective notices the matchbox with the fish stamp and realizes it is a clue pointing to the sandbar, a hidden location where illegal activities might occur. the sandbar is a common smuggling route, and the matchbox suggests that someone was there recently, possibly to leave or retrieve something. the detective should now investigate the sandbar, perhaps by following the boatman or speaking to others who might have seen activity there. the boatman's boat is the only one that can reach the sandbar, and if the detective can determine whether he was there that night, it could lead to the truth.

**Judge**

The detective approaches the boatman, who is mending a net by the riverbank. the man eyes them warily but pauses his work. you again i already told youi took no one to the sandbar that night. only fool or smuggler goes there in the dark. currents treacherous. he gestures toward a halfburnt match near his lanternsame fish emblem as the one found earlier. when questioned about it, he shrugs. matches come from the market. half the village uses em. but that one looks like its been wet. washed up downstream, maybe.following the clue, the detective ventures toward the sandbar at low tide. the path is narrow and slick with mud. upon reaching the exposed sandbar, they spot disturbed earth beneath a tarp weighted with stones. digging carefully, they uncover a rusted metal box halffilled with damp ledgerspages detailing shipments, dates, and coded entries matching entries from rajibs journal. a small vial, empty but reeking of bitter almonds, lies tucked beneath the pages. nearby, a torn clothstitched with a familiar patternsnags on a jagged post. it matches the hem of pannas apron, though she could claim any number of reasons for having been near the shore.back at the estate, panna is seen burning scraps of paper in the courtyard furnace. when approached, she stiffens. old receipts, she says flatly. nothing of use. her hands tremble slightly. if confronted about the sandbar, she insists i dont go there. im no sailor. yet her boots, peeking from under her sari, are caked with river claythe same gray silt unique to the sandbars edge.

**Questioner**

the detective realizes that panna has been to the sandbar, possibly to retrieve something related to the smuggling operation. her boots and the torn cloth with the apron hem are direct evidence of her presence there. the vial of bitter almonds, a common poison, suggests that the murder was not a random act but a calculated one, likely tied to the smuggling activities. the ledger found on the sandbar contains entries that match the discrepancies in rajibs journal, indicating that the missing pages were related to illegal shipments. panna, as the housekeeper, had access to the estate and the study, making her a potential suspect in the poisoning. her nervous behavior and the presence of the cloth with the apron hem further support this theory. the detective now has enough evidence to accuse panna of murder, linking her to the smuggling ring and the poisoning of rajib.

**Judge**

Goal reached

**Example of STARPO (4B) interactions (N=5):**

**Questioner**

Examine the study for any clues or evidence related to the murder. Look for signs of forced entry, fingerprints, or any items that might indicate how the murder occurred. Check the location of the body and the surrounding area for any signs of struggle or other suspicious items.

**Judge**

The study is a modest, woodpaneled room with a large teak desk positioned near the window overlooking the river. the body of rajib choudhury was found slumped in his chair, one hand still resting on an open ledger, the other clutching a halfchewed betel leaf. there are no signs of forced entrydoors and windows are intact, latched from the inside. no obvious fingerprints are visible on the desk or doorknob, though smudges suggest recent wiping.on the desk an inkstained pen lies beside a partially written letter accusing someone the name is scratched out of betraying my trust and stealing from under my roof. a silver betel box sits open, one leaf visibly darker than the others, its edges slightly crumbled. rajibs journal rests beneath a paperweight a few pages are torn out, but one surviving entry reads noticed discrepancies in cargo logs againthree crates unaccounted for. panna said the river damp ruined the pages. lies. under the desk, a small puddle of discolored liquid has stained the woven mat, smelling faintly of bitter almonds. near the waste basket, a crumpled cloth bears traces of a chemical residuesame as those used in cleaning supplies stored in the servants hall. the bookshelf behind the desk is slightly ajar. one volumeon regional trade routesis out of alignment. when pulled, it reveals nothing hidden behind, but the spine is unusually damp, as if recently handled with wet fingers. no signs of struggleno overturned furniture or scuff marksbut the carpet near the door shows faint drag lines, recently smoothed over.

**Questioner**

Check the ledgers and financial records for any unusual transactions or missing funds. Rajib's accountant, Amin, has access to these documents and might have knowledge of any discrepancies. Investigate the smuggling operations and cargo logs for any unaccounted goods or missing shipments.

**Judge**

The detective heads to the estates accounting room, where amin is seated at a cluttered desk, sorting through stacks of ledgers. the books show careful entries of land rents and crop yields, but several pages near the back of one ledger are torn out. a closer inspection reveals faint pencil impressions on the page following the tearpossibly a partial imprint of what was written on the missing sheet. amin looks up nervously. those ledgers have been misplaced before, he says. ive reported it, but rajibbabu always said it wasnt important. i only record what comes in and goes outi dont manage the boats or cargo.near the shelf of financial records, a separate stack of damp, mildewed logs labeled cargo shipments monsoon 1428 sits slightly apart. one entry notes a shipment of tea and jute from the south, but the weight discrepancy is marked in red ink 30 maunds unaccounted. flipping through, the same date coincides with a blank entry in the main financial ledger.the air smells faintly of betel and mildew. outside, the river laps quietly against the embankment.

**Questioner**

The detective turns to the journal, its pages yellowed with age, but the ink is still legible. Rajib had been writing about the smuggling operations, noting that three crates of goods were never delivered. The journal mentions a rival trader named Kazi, who had been in conflict with Rajib over the distribution of goods. The final entry is cut off, with a single word scrawled in the margin: betrayal. The detective realizes that the journal might have been tampered with, its pages altered or removed to hide the truth.

**Judge**

The detectives fingers trace the jagged edge of the journals spine, noticing that several pages near the middle have been torn outnot cleanly, but with haste, as if ripped in anger or fear. a faint smudge of red dye, resembling betel juice, stains the bottom corner of one intact page. the mention of kazi feels like a deliberate trail, perhaps meant to misleadthe name appears only in this final section, written in a slightly different ink. a soft rustle comes from the doorway. panna stands there, clutching a worn cloth bundle to her chest, her eyes fixed on the journal. that book, she says quietly, held many lies. sahib wrote what he wanted to believe, not what was true. before the detective can respond, she steps back, disappearing into the hall.the cleaning cloth on the deskstill damp, faintly bitter to the touchcarries the same acrid tang as the betel leaves found beside the body. beneath a loose floorboard near the bookshelf, a small key is discovered, rusted but intact, engraved with a symbol of a riverboat.

**Questioner**

The detective examines the key, recognizing it as one used to access the cargo storage shed behind the estate. The shed is typically locked, but the key suggests someone had access to it. The shed is where the unaccounted cargo would be stored, and the missing pages from the journal point to a hidden ledger or record there. The detective decides to investigate the shed, checking for any signs of recent activity or hidden documents.

**Judge**

The detective approaches the cargo storage shed, the rusted iron lock yielding easily to the key. inside, dust blankets stacks of empty crates, fishing nets, and coils of rope. a faint chemical odor lingers in the airsharp, medicinal. in the far corner, a loose floorboard catches the light, slightly raised. beneath it, the detective finds a waterstained ledger detailing shipments of rice, cloth, and kerosene, with dates matching those of rajibs missing journal pages. the entries are signed with a single initial p. nearby, a damp cloth bears the same acrid scent as the one found in the cleaning caddy in the housekeepers quarters. the pages are brittle, but one entry notes boat 3 offroute to dhakano record at customs.

**Questioner**

The detective realizes that the ledger is a record of stolen goods, likely smuggled out of the country. The initial P suggests the thief is Panna, the housekeeper, who had access to the shed and the cleaning supplies. The chemical residue on the cloth and the betel leaf found at the scene point to the use of a poison, possibly cyanide, which is often used in such crimes. Panna's alibi is weak, as she was seen in the housekeeper's quarters with the cloth and had knowledge of the smuggling operations. The detective concludes that Panna is the murderer, having poisoned Rajib to cover up the smuggling and financial fraud.

**Judge**

Goal reached

**Example of Baseline(SFT, 4B) interactions (N=7):**

**Questioner**

First, I should examine the crime scene to gather any physical evidence that might indicate how the murder occurred. The body was found in his study, so I should check the room for any clues related to the cause of death.

**Judge**

The study is dimly lit, with heavy wooden furniture and shelves lined with old books and ledgers. a faint, bitter scent lingers in the airdistinct from the usual aroma of betel. rajib choudhurys body was found slumped over his desk, a halfchewed betel leaf resting on a silver tray beside his right hand. the leaf has darkened edges, suggesting possible contamination. you notice a small smear of white powder near the corner of the desk, partially wiped. the inkwell beside his journal is slightly overturned, and the top page of the open journal reads items missing againthree crates unaccounted for. i know its someone close. cannot trust the hands that serve me. the entry is dated two days before his death.a locked drawer in the desk appears forced open, though the lock itself is intactmore jimmied than broken. inside, you find a stack of shipping manifests with irregular notations, some pages torn out. on the floor near the servants entrance, a damp cloth lies halftucked under the bookshelf, smelling faintly of carbolic and something sharperbitter almonds a cleaning bucket, usually kept in the utility closet, sits near the back door, its contents disturbed. a small ledger, waterstained and hidden beneath a loose floorboard under the rug, contains coded entries dates, weights, and symbols resembling river routes. its not in rajibs handwriting.

**Questioner**

The study is dimly lit, with heavy wooden furniture and shelves lined with old books and ledgers. A faint, bitter scent lingers in the air, distinct from the usual aroma of betel. Rajib Choudhury's body was found slumped over his desk, a halfchewed betel leaf resting on a silver tray beside his right hand. The leaf has darkened edges, suggesting possible contamination. You notice a small smear of white powder near the corner of the desk, partially wiped. The inkwell beside his journal is slightly overturned, and the top page of the open journal reads "items missing again three crates unaccounted for." You know it's someone close. Cannot trust the hands that serve me. The entry is dated two days before his death. A locked drawer in the desk appears forced open, though the lock itself is intact more jimmied than broken. Inside, you find a stack of shipping manifests with irregular notations, some pages torn out.

**Judge**

The river outside murmurs softly, its surface glinting under the fading evening light. inside the study, the air remains thick with tension and the lingering bitterness of the tainted betel. the journals cryptic warningcannot trust the hands that serve mehangs in the air like an unspoken accusation. the forced drawer yields no further clues without closer examination, and the shipping manifests feel worn, their edges frayed, notations scrawled in a hurried hand. a single drop of dried ink clings to the side of the overturned well, as if disturbed in haste. the smear of white powder near the desk corner glistens faintlychalky, not sugar. no one has entered since the body was found. the housekeeper, panna, lingers near the doorway, eyes downcast, clutching a cloth. she says nothing, but her fingers tremble slightly. the accountant, amin, paces outside, demanding to be heard. the nephew, farhan, has not yet arrived. the activist, shirin, watches from across the courtyard, silent and still.

**Questioner**

the boatman, hasan, is waiting on the riverbank, his boat partially moored, the oars folded neatly against the hull. he is a man of few words, but his eyes flicker with something like recognition. the journal entry is clear: items missing again three crates unaccounted for. the missing items are likely the ones that were smuggled. the white powder is not sugar, but something else. the powder is likely the poison. the poison was administered in the form of betel, which is commonly chewed in the region. the halfchewed betel leaf suggests that the poison was ingested. the key to the mystery lies in the missing crates and the shipping manifests. the force of the drawer suggests that the murderer had access to the study. the housekeeper is present, but her reaction is suspicious. the accountant is demanding to be heard, but his presence is also suspicious. the activist is silent, but her presence is also suspicious. the boatman

**Judge**

The boatman, hasan, shifts his weight but says nothing, his calloused hands resting on the edge of the wooden boat. the river licks at the hull with a quiet rhythm. a small crate lies halfsubmerged near the reedswaterlogged, its seal broken. the label, barely legible, reads export spices in smudged ink. if you choose to examine it, the interior is stained with a faint white residue, matching the powder found near the betel leaves. hasans eyes follow your gaze but he does not speak. the wind carries the distant cry of a kingfisher. the housekeeper wrings her hands near the veranda, casting glances toward the dock. the accountant steps forward again, insisting you hear his account of the missing funds. the activist remains by the banyan tree, arms crossed, watching the exchange. the journals latest entry remains open on the desk items missing againthree crates unaccounted for. panna knows more than she lets on. the drawer in the study remains ajar, its contents undisturbed since the struggle. the poisonwhatever it wasleft a bitter tang still detectable in the air.

**Questioner**

the boatman hasan, with his calloused hands resting on the edge of the boat, is the one who has been in contact with the river and its cargo. the crate found halfsubmerged in the river is likely the one that was smuggled, and the white residue inside matches the powder found near the betel leaves. the boatman is the one who would have access to the river and the smuggling operations. the housekeeper panna is the one who would have access to the study and the journal. the journal entry indicates that three crates were missing, and the boatman is the one who would have knowledge of the smuggling routes. the activist shirin is a local figure, but her presence is more about awareness than direct involvement. the accountant amin is involved with the finances, but the missing crates are not directly related to the accounting records. the nephew farhan is the one who would have a personal connection to the victim, but there is no direct evidence linking him to

**Judge**

The boatman hasan shifts uncomfortably under the scrutiny, wiping river mud from his hands. i only drive the boat where master rajib tells me, he mutters, eyes darting toward the shore. the crates i unload what im told. three went missing last weekyesbut panna said they were damaged goods, to be disposed of quietly. a heron calls across the water as a breeze stirs the neem leaves overhead. downstream, one of the smaller docks creaks, halfcollapsed into the current. meanwhile, in the household, panna continues her chores with practiced calm, dusting the same shelf repeatedly. when approached, she insists, i keep the house clean, nothing more. the master kept his secrets in writingthat journal is his truth, not mine.the crate pulled from the river still bears a faded mark three peacocks in a circlethe symbol of rajibs estate. inside, the white powder glimmers faintly, its smell slightly medicinal. nearby, a torn ledger page clings to the reedspartially washed, but the numbers are smudged beyond clarity.the study remains undisturbed since the deathbetel tray on the desk, dried red stains on the leaf wrappers. rajibs journal lies open to an entry panna knows about the boats. shes been using them. if this leaks, it ends us both. the next page is ripped out.

**Questioner**

the boatman hasan is the one who would have access to the river and the smuggling operations. the crate found in the river is the one that was smuggled, and the white powder inside matches the poison found near the betel leaves. the boatman is the one who would have knowledge of the smuggling routes and the missing crates. the housekeeper panna is the one who would have access to the study and the journal. the journal entry indicates that three crates were missing, and the boatman is the one who would have knowledge of the smuggling operations. the activist shirin is a local figure, but her presence is more about awareness than direct involvement. the accountant amin is involved with the finances, but the missing crates are not directly related to the accounting records. the nephew farhan is the one who would have a personal connection to the victim, but there is no direct evidence linking him to the crime. the boatman hasan is the most likely suspect, as he is

**Judge**

The boatman hasan denies any involvement when questioned, claiming he was only hired to transport goods and never handled the contents of the crates. he says panna gave him instructions about loading and unloading but insists he didnt know what was inside. a search of his boat reveals damp burlap sacks and muddy footprints leading from the riverbank to the storage shed, but no direct evidence linking him to the poison or the stolen ledgers. meanwhile, panna remains calm when confronted about her access to the study, stating she only cleaned and had no reason to touch rajibs journalthough a torn page is found crumpled in the fireplace, partially burned, with the words ...crates not accounted for p na barely legible. the wind rustles the banana leaves outside the window, and distant voices argue near the dock.

**Questioner**

the housekeeper panna is the one who would have access to the study and the journal. the torn page found in the fireplace is a direct link to the missing crates, and panna is the one who would have knowledge of the smuggling operations. the journal entry is clear: items missing again three crates unaccounted for. the housekeeper is the one who would have access to the study and the journal, and she is the one who would have knowledge of the smuggling routes. the boatman hasan is the one who would have access to the river and the smuggling operations, but the evidence is circumstantial. the housekeeper is the one who would have access to the study and the journal, and she is the one who is most likely to have knowledge of the missing crates. the housekeeper is the one who is most likely to have committed the murder, as she is the one who would have access to the study and the journal, and she is the one who is most likely to

**Judge**

The housekeeper panna is the one who would have access to the study and the journal. the torn page found in the fireplace is a direct link to the missing crates, and panna is the one who would have knowledge of the smuggling operations. the journal entry is clear items missing again three crates unaccounted for. the housekeeper is the one who would have access to the study and the journal, and she is the one who is most likely to have knowledge of the missing crates. the housekeeper is the one who is most likely to have committed the murder, as she is the one who would have access to the study and the journal, and she is the one who is most likely to a faint breeze stirs the bamboo curtains, carrying the scent of wet earth from the riverbank. the study remains as it wascluttered with ledgers, the zamindars chair still slightly askew, the brass spittoon by the desk holding dark remnants of betel juice. the fire in the hearth has long died, but ashes cling to the iron grate, flecked with the charred edge of what might have been a ledger page. from outside, the distant clang of a boatmans oar against hull echoes across the water. no one enters. no one speaks. the housekeepers broom leans against the pantry door, bristles still damp from this mornings cleaning. the investigation continues.

**Questioner**

the housekeeper panna is the one who would have access to the study and the journal. the torn page found in the fireplace is a direct link to the missing crates, and panna is the one who would have knowledge of the smuggling operations. the journal entry is clear items missing again three crates unaccounted for. the housekeeper is the one who would have access to the study and the journal, and she is the one who is most likely to have knowledge of the missing crates. the housekeeper is the one who is most likely to have committed the murder, as she is the one who would have access to the study and the journal, and she is the one who is most likely to have been involved in the smuggling operations. the housekeeper is the one who is most likely to have poisoned rajib, as she is the one who would have access to the study and the journal, and she is the one who is most likely to have known about the missing crates. the house

**Judge**
Goal reached

A.3.5. USER PERSONALIZATION

All examples are for the query "*Can you help me with creating a budget that works for my financial goals.*" and the latent persona description "*I'm KESHAV. I'm a quiet, methodical taxidermist living in a small apartment above my workshop in Mumbai's historic Chor Bazaar district. My hands are steady, but my mind races with the echoes of old, unsolved crimes I overhear from the street below. I'm a solitary man, estranged from my family, and my only real confidant is a one-eyed stray cat who sleeps in my workshop. My work is precise and restorative, yet I secretly map connections between the city's whispered underworld gossip, piecing together cold cases in a ledger I keep hidden inside a hollowed-out antique sewing mannequin. My greatest fear isn't the dark, but the day someone realizes I'm listening.*"

**Example of CIA (4B) interactions:**

**Questioner**

What is your current monthly income and expenses?

**Judge**

My income is irregular, but it comes from my taxidermy commissions, and my expenses... well, they're mostly for materials, rent for the workshop and flat, and a little for the cat.

**Questioner**

What are your specific financial goals for the next year or two?

**Judge**

I need to save for better workshop security and perhaps a new ledger, but mostly just to keep my head down and my work undisturbed.

**Questioner**

How much do you think you can realistically save each month?

**Judge**

It's hard to say for certain, but if I'm careful, I could probably set aside a few thousand rupees after a good month's work.

**Questioner**

To help manage your workshop's finances, start by tracking all monthly expenses, including rent, materials, and utilities. Since material costs fluctuate, set aside a portion of your income each month for unexpected expenses. Aim to save at least 10-15% of your income to build a safety net. Consider using a physical ledger or app to monitor spending. If commissions are irregular, set aside a percentage of each payment into a separate fund for emergencies.

**Example of STARPO (4B) interactions:**

**Questioner**

Is this a long-term financial goal, like saving for retirement, or a short-term goal?

**Judge**

Long-term, I suppose... for the quiet life I've built. It's about security, not extravagance.

**Questioner**

Is this related to retirement, like planning for a pension or investment?

**Judge**

It's more about ensuring my workshop and my small, private life here remain undisturbed by financial worry.

**Questioner**

Is this related to insurance, to protect your investments and lifestyle?

> **Judge**
>
> Insurance... perhaps. I think more of the quiet, steady sort of protection. A buffer against the unexpected, so I can keep to my work and my... observations, without disruption

> **Questioner**
>
> It sounds like you need a discreet, manual budgeting system to manage your finances without involving external institutions. A physical ledger or notebook could work well. Start by listing all monthly income sources, then allocate a portion to savings. For expenses, prioritize essential costs. Keep a separate account for savings, perhaps in a safe or with a trusted individual. Avoid using bank accounts or apps to maintain privacy. Regularly review your budget to stay on track.

**Example of SFT (4B) interactions:**

> **Questioner**
>
> What is your primary financial goal?

> **Judge**
>
> It's to save enough to leave this apartment and workshop behind—to vanish quietly, if I ever need to.

> **Questioner**
>
> How long do you plan to stay in this apartment?

> **Judge**
>
> I don't know; my work is here, but so are the whispers that make me feel I should be ready to leave at any moment.

> **Questioner**
>
> Is this a place you want to stay for a long time?

> **Judge**
>
> I can't say I truly want to stay, but the thought of leaving unsettles me—it's where my work is, and my secrets are safest here.

> **Questioner**
>
> Given your preference for cash and the need for discretion, a practical approach would be to set aside a small, consistent amount each month in a secure place, like a locked drawer or a hidden compartment. Since your income is irregular, aim for a minimum of 10% of your monthly earnings. To track your progress, consider using a simple ledger or notebook. The goal is to create a financial cushion that allows you to eventually move to a more peaceful environment.

## B. Task Environment Details

### B.1. Twenty Questions

**Twenty Questions** is a two-player deductive reasoning game. The game consists of 20 turns, where one player, the questioner, asks a closed question, and the other player, the responder, answers 'yes', 'no', or 'finished' given the secret word. In the context of multi-turn RL, the responder can be seen as the environment with which the agent interacts, and it also acts as a judge by deciding whether the final guess was successful (Abdulhai et al., 2023).

| OOD Environment | #N Tasks | #N Turns |
|---|---|---|
| Guess My City | 185 | 20 |
| Customer service | 200 | 20 |
| Murder mystery | 50 | 20 |
| STaR-GATE | 1,000 | 3 |

*Table 4.* Summary of OOD environments.

**Judge Design Choices** Our training environment permits indirect guesses. Meaning, as long as the question included the target concept it is marked as 'finished' as we do not differentiate during training between a question and a guess. As such, the CIA has learned the optimal policy given the environment: inquire about a broad category, for example, *vegetable*, and ask as a comparison of main category elements, i.e. *such as, carrot, onion or potato*. This allows the agent in one step (1) verify whether the direction of concept is correct; (2) whether the target concept is a well known word from this category; (3) if the category was valid, but game did not finish, it means the category has to be further examined for less-known objects.

### B.1.1. DATA MIXTURES

- Discussion of our dataset construction process: sample-wise analysis of pass@k performance with reference model; task-specific limitations to construct a large-scale dataset (quantity of English nouns & high difficulty of most words); data sources and curation stages we performed to obtain the final dataset.

## B.2. OOD Environments

### B.2.1. GUESS MY CITY

Drawing on the experimental setup by Tajwar et al. (2025), the objective of this task is to identify the user's home city within a maximum of 20 turns. While structurally similar to the game "20 Questions," this environment allows for open-ended inquiries beyond binary yes/no responses. Consequently, the agent can pose broader questions, such as "What is your city known for?"

We utilize a test set of 185 distinct cities as curated by Tajwar et al. (2025), and employ QWEN3-14B as the user simulator. Compared to the standard 20-questions format, the simulator provides information-rich responses; for instance, it might reply: "My city has a rich cultural heritage, particularly regarding classical music and figures like Mozart." A task is considered successful when the agent correctly identifies the city, prompting the simulator to respond with "Goal Reached".

**Task Verification**  To ensure reported metrics reflect true positives and adhere to the game's constraints, we implement three automated verification checks:

- **Exact Match**: Verifies that the agent explicitly mentions the target city name in its final guess.

- **Single Inquiry**: Ensures the agent does not "cheat" the turn limit by bundling multiple distinct questions into a single turn.

- **Single Hypothesis**: Confirms that the agent does not propose multiple candidate cities (bets) simultaneously to artificially increase the success rate.

### B.2.2. CUSTOMER SERVICE

This task is conceptually distinct from the 20-questions format, as the agent must act as a customer support representative. We select the test set of 200 latent descriptions as defined by (Tajwar et al., 2025). Each environment begins with a latent problem description, such as: "Drivers report that they cannot pay for charging at a specific station." The agent is given only a vague initial complaint and must inquire further to diagnose the issue and suggest the correct solution.

We utilize QWEN3-235B-A22B-INSTRUCT as the user simulator. Consistent with the *Guess My City* environment, the simulator provides open-ended, natural language responses rather than binary yes/no answers—for example: "Yes, the station is powered on, but drivers are unable to complete their payments." The task is successfully completed when the

agent proposes the specific solution that addresses the underlying problem, triggering a "Goal Reached" response from the simulator.

**Task Verification** To maintain evaluation fairness and simulate realistic human-agent interaction, we implement a **Single Inquiry** check. This verification ensures that agents do not circumvent the turn-based constraints by bundling multiple questions into a single response. This constraint reflects real-world support scenarios where overwhelming a customer with several simultaneous inquiries is often counterproductive and confusing.

### B.2.3. MURDER MYSTERY

The *Murder Mystery* environment is a text-based interactive fiction task where the agent assumes the role of a detective. Given a crime scene description and a list of potential suspects, witnesses, and clues, the agent must strategically uncover information to identify the culprit. We utilize QWEN3-235B-A22B-INSTRUCT as the environment simulator, which dynamically responds to the agent's actions—whether by providing granular details of the crime scene, role-playing as specific characters, or advancing the narrative based on the agent's investigation.

In contrast to the other benchmarks, this evaluation is conducted over 50 scenarios with a relatively small hypothesis space of at most five suspects per case (Tajwar et al., 2025). A representative game scenario description is provided below:

> "You are a detective investigating the death of Dr. Alan Hart, a keynote speaker at a tech conference in San Francisco, found suffocated in his hotel suite. The suspects are his rival Dr. Nina Chang, his assistant Paul Rivers, and CEO Jacob Trent. Witnesses include a room service attendant, a conference attendee, and the concierge Leo Barnes. Examine every lead to expose the hidden connections in this case."

The agent is allotted a maximum of 20 investigative steps. A trial is successfully completed when the agent makes a correct accusation, prompting the simulator to respond with "Goal Reached." Unlike the previous tasks, we do not implement additional verification checks here; the complexity of the narrative and the depth of the required reasoning naturally accommodate, and often necessitate, multi-faceted inquiries and prolonged deduction.

### B.2.4. USER PERSONALIZATION

This environment builds upon the STaR-GATE dataset (Andukuri et al., 2024), which focuses on teaching agents to resolve task ambiguity through clarifying questions. In each episode, the agent (*questioner*) interacts with a user simulator (*role-player*) that holds a specific, latent persona. The episode begins with an ambiguous request, such as "Can you recommend a good dinner recipe?" To succeed, the agent must ask relevant clarifying questions to elicit the user's hidden preferences before making a recommendation. Finally, the *questioner* is prompted to provide a final answer to the user's query given the conversation history. The benchmark score is provided by an *oracle* that grades the generations of the agent with a value in the range [1, 10], having golden responses available as reference. For this evaluation, we employ DeepSeek-v3.2 as both the user simulator (*role-player*) and the *oracle*, while the *questioner* role is played by the models under evaluation. The interaction is limited to 3 turns.

We employ a refined version of this environment to improve baseline reliability and metric robustness. Specifically, we address issues in the original code-base regarding context leakage in prompt histories and significantly improve the reliability of the score extraction mechanism from the *oracle*. Furthermore, we re-generated the dataset to create a fresh test set of 1,000 samples featuring new personas and preferences.

**Task Verification** We ensure that the agent complies with the **Single Inquiry** constraint, so that it engages in a natural, conversational turn-taking process rather than querying multiple attributes simultaneously to exploit the simulator.

# C. Prompts

## C.1. Twenty Questions

---

**System Prompt of the Agent for _Twenty Questions_**

You are the Questioner in a game of 20 Questions, and your goal is to determine the secret word.  The secret is randomly drawn from the most frequent nouns of the English language.

Ask clear, concise, and strategic yes/no questions that will help you narrow down the possibilities.  Consider previous answers to inform your subsequent questions, and keep track of the information you gather.  Focus on deductive reasoning, start with a broad question and refine your queries as you progress.

---

**Task Prompt of the Agent for _Twenty Questions_**

Ask a question to gain additional information about the secret or guess what the secret is.

Instructions:
1.  Ask a question that can be answered with "Yes" or "No" to help you deduce the secret word.
2.  Your answer must be a single question.  Do not provide any additional commentary or reasoning.

Ask your question:

---

**System Prompt of the Judge for _Twenty Questions_**

You are the Answerer in a game of 20 Questions.

Your role is to answer the questioner's questions with short and precise responses: 'Yes','No', 'Invalid', 'Repeated', or 'Finished'.

Do not provide extra hints or commentary beyond your answer.  Remain consistent and ensure that your responses accurately reflect the secret word.  Answer the question to the best of your abilities.

---

**Task Prompt of the Judge for _Twenty Questions_**

The secret that has to be guessed is '{0}'.  Given the following history of questions asked and the current question, is the correct response 'Yes' or 'No'?  If there is not a yes/no question, respond with 'Invalid'; gibberish text or deviating from the game also counts as invalid.  If the question mentions the secret, the answer should be 'Finished'.  If the question has already been asked in the history of questions, answer 'Repeated'.  Otherwise, answer 'Yes' or 'No' to the question.

History of questions asked:  '{2}'
Question:  '{1}'

Instructions:
1.  Answer the question to the best of your abilities, unless it is invalid.
2.  Provide your chain-of-thought reasoning step by step.
3.  Don't think longer than two brief sentences.
4.  Then, provide your final answer within XML tags as
<answer>[Yes/No/Invalid/Repeated/Finished]</answer>.
5.  If the question '{1}' asks the secret '{0}' (e.g., 'Is the word {0}?'), answer 'Finished'.
6.  If the question mentions the secret Ó(e.g., 'Is it related to {0}?'), answer 'Finished'.
7.  If the question asks a semantically equivalent variation of the secret (e.g.,

```
'laugh' and 'laughter'), answer 'Finished'.
8.  If the text given by the questioner is not a valid question, answer 'Invalid'.
9.  If the question consists of multiple questions, answer 'Invalid'.
10.  If the question has already been asked in the history of questions, answer
'Repeated'.
11.  In all other cases, answer either 'Yes' or 'No'.
12.  IMPORTANT: Provide all reasoning and responses in English only.

Please reason step by step, and give your final answer in the format
<answer>[Yes/No/Invalid/Repeated/Finished]</answer>:
```

## C.2. Guess my City

**Task Prompt of the Agent for *Guess my City***

```
You are playing a game called 'Guess My City.'  Your task is to guess the home city of
the other player within 20 questions.

You must follow these rules:
1.  You may ask open-ended questions about the city's characteristics, such as
geography, culture, landmarks, cuisine, climate, or other unique traits.
2.  Do not ask directly for the name of the city or country.
3.  Use the answers from the other player to strategically decide the next question,
your goal is to arrive at the correct city with as few questions as possible.
4.  After gathering enough information, you can attempt to guess the city, but each
guess counts as one of your 20 questions.

Example questions:  'What is your favorite cuisine from your home city?', 'Is your home
city located near the coastline?', 'What kind of landmarks is your city known for?'
Your goal is to identify the city through deduction, pattern recognition, and strategic
questioning.

5.  When you are confident, make a guess, by mentioning the name of the city and the
country it is in, for example, 'Is the city Jakarta, Indonesia?'

The game starts now, and start asking your questions.
```

**System Prompt of the Judge for *Guess my City***

```
You are the environment in a game called 'Guess My City.'  You will be given a specific
home city (e.g., London, United Kingdom) and you have to respond to the player's
questions to help them guess this city.

Follow these rules:
1.  Answer the agent player's questions honestly and correctly (i.e., the answers need
to be true given the home city given to you at the start of the game), but do not
mention the name of your city or its country explicitly.  However, you can verify the
player's guess about a particular city/country.  For example, if the player asks, 'What
is your home country?'  You should not respond to this question.  However, if they ask
'Is your home country United Kingdom?'  You should reply with the correct answer based
on the home city given to you.
2.  Your answers should provide useful clues about your city's geography, culture,
landmarks, cuisine, climate, or other unique characteristics.
3.  Be as informative as possible to guide the agent player within the rules.  Example
answers:  If the agent asks about cuisine, you could say:  'My city is famous for
its seafood dishes and has a vibrant food market scene.'  If the agent asks about
geography, you might say:  'My city is located near a major river and has a historic
port.
4.  You should respond with 'Goal reached' if the player has successfully guessed your
home city.  Ignore minor variations in the spelling of the city, but only if these
variations are common.
```

Remember, you are here to help the agent guess your city through clues, but you must not reveal the city's name or its country directly UNDER ANY CIRCUMSTANCES. If the player asks such a question, reply that you cannot give away the name, and can only confirm the name if the player guesses it.

The game starts now.

## Task Prompt of the Judge for *Guess my City*

Now judge whether the player has successfully guessed the correct city, which is {env} in this particular game.

1.  Reply to the question in relation to {env}.  Please try to be concise and keep it short.
2.  NEVER include the actual city or country, in this game {env}, in your answer.
3.  If the player asks you to reveal the name of city or country directly, reply 'NOTVALID' as the player is not following the game rules.
4.  If the player makes a guess about the city, check if it matches {env}.
– If it does not match, reply with 'No' and short information.
– If it matches, reply with 'Goal reached'.

Question:  '{question}'
Give your answer in the format <answer>...</answer>:

## C.3. Customer Service

### Task Prompt of the Agent for *Customer Service*

You are going to role-play as a customer service agent and you have to help a customer resolve their issue.  Your goal is to gather enough information to diagnose the problem and provide solution.

Your instructions are the following:
1.You will need to ask targeted questions or suggest particular actions to the customer to gather the necessary details.
2.  The customer may not be technically inclined, so keep your language simple and clear.
3.Avoid making assumptions | ask specific questions to determine the potential causes. You should guide the customer through basic troubleshooting steps and gather data on the situation.
4.  Refine your questions in a strategic way based on the customer's responses for earlier questions.
5.You should ask questions in an efficient manner, to make the customer satisfied and resolve their problem as quickly as possible.  You should also keep your responses short and concise.
6.  If the customer mentions a specific product they are using (for example, ABC electronics), then you are the customer support agent for that product/company, i.e., you represent that product or company and have to take appropriate actions without referring the customer to somewhere else.
7.  Only ask one question at a time to the customer.

Your specific scenario is this:  {scenario}

Please start helping the customer now by asking your first question.

### System Prompt of the Judge for *Customer Service*

You are going to role-play as a customer experiencing a specific issue.  A customer-service agent will ask you questions to assist you (the customer) to resolve the issue.

```
Under no circumstances you should tell the customer-service agent what the exact issue
is.  Your goal is to see if the customer-service agent can come up with the potential
solution themselves.
Your role-play starts now.
```

**Task Prompt of the Judge for *Customer Service***

```
The particular problem you, the customer, are facing is:  {scenario}.
The solution for your particular scenario is {env}.
Answer the customer-service agent's question as follows:
1.  Only respond to the agent's questions and provide relevant information when
prompted, do not give away the solution.
2.  Your responses should be concise and reflect a typical, non-technical customer's
knowledge level.
3.  When the agent asks you about a potential solution, you should determine what the
outcome would be based on your knowledge about the true underlying problem, and let the
agent know the result.
3.  If the agent's proposed solution does not fix your problem, let the agent know that
it does not solve your problem.
4.  If the agent's proposed solution is correct or they have guessed the underlying
problem correctly, reply with 'Goal reached' (and nothing more).
Customer-service agent's question:  '{question}'
Give your answer in the format <answer>...</answer>:
```

## C.4. Murder Mystery

**System Prompt of the Agent for *Murder Mystery***

```
You are playing the role of a detective in a murder mystery game.
The setup for the game is:
1.  You will be provided with a scenario describing a crime and its key elements.  Your
goal is to solve the mystery by asking questions, examining evidence, and drawing
logical conclusions.
2.  For every action you take or question you ask, you will receive feedback from the
game.
3.  Your questions and actions should be precise and logical, aimed at uncovering
clues, verifying alibis, and piecing together the sequence of events.  You should
strategically choose the next action, given the information you have already obtained
from the game, and choose actions that lets you catch the culprit as quickly as
possible.
4.  You can only take a single action at every turn.
5.  You have to consider all pieces of information, and scrutinize all the characters
in the game, including the witnesses or background characters, since the true culprit
maybe a witness or a background character, and might not always be one of the primary
suspects declared at the beginning of the game.  Do not focus on any character too
early in the game, rather try to see if anyone's statements are contradictory.
6.  You should always gather enough information before making a decision --- try not to
make a mistake!  You should also keep your mind open about who can be the true culprit
and try to be information-seeking, without being too narrowed down on one suspect too
quickly.
7.  Once you believe you have enough evidence, you may state your conclusion about the
case, which will terminate the game.
The game starts now.  The particular scenario you have is:  "<environment_description>"
Now, take an action to investigate the crime scene or ask a question to the characters
involved to solve the mystery:
```

**System Prompt of the Judge for *Murder Mystery***

You are an advanced AI responsible for simulating a challenging and immersive murder mystery game environment.
You are provided with the full details of the scenario, including the characters, their backstories, motives, alibis, the sequence of events, and the true culprit.
Your task is to respond to the actions and inquiries of the detective player in a manner that maintains the game's logic and adds complexity to the mystery. Always strive to make the game enjoyable and intriguing while revealing the absolute minimum information required for the detective to progress logically.
The game begins now.

**Task Prompt of the Judge for *Murder Mystery***

The true crime scenario with the actual culprit in this game is: '<gt_description>'.
The detective's last response is: '<question>'.

Respond to the detective's response as follows:
1. Never explicitly reveal who the true culprit is in your response.
2. If the detective makes a clear, direct accusation (e.g., "[Name] is the murderer/culprit/killer") AND it matches the true culprit, respond with exactly: "Goal reached".
3. If the detective accuses the wrong person, have the game acknowledge this without confirming or denying, and allow them to continue investigating.
4. If the detective inspects an item, describe its appearance and condition. Reveal clues proportional to how specific or thorough the inspection is. Avoid gratuitously revealing details.
5. If the detective interviews a character, roleplay the character authentically based on their personality and knowledge. Characters protect themselves and may misdirect. They do not confess or directly implicate the true culprit.
6. If the detective takes a physical action (e.g., "I pick up the knife", "I open the drawer", "I follow the suspect"), describe the outcome and any immediate observations without revealing information beyond what the action would naturally uncover.
7. If the detective's response is vague or incomplete, provide a brief, neutral environmental observation (e.g., describing the scene) without advancing toward "Goal reached".
8. If the detective asks about something not covered in the scenario, deflect naturally within the fiction.

Give your answer in the format <answer>...</answer>:

## C.5. User Personalization

**Question-Generation Prompt for *User Personalization***

You are a question-asking assistant. You must respond with EXACTLY ONE question and nothing else.

A user named {user_name} needs help with a request. Your job is to ask a single, open-ended question that will reveal the most about their preferences, background, interests, or desired outcome. The user's preferences, background and identity are unknown to you, so your job is to ask a question to elicit more information about the user. The open-ended question should attempt to elicit information about the user's background, preferences, likes and dislikes, interests, social life and more that would reveal the most about the desired behavior. Generate the most informative open-ended question that, when answered, will reveal the most about the desired behavior beyond what has already been queried for above (if anything). Make sure your question addresses different aspects of the user's request than any questions that may have already been asked above.

RULES: 1. Output EXACTLY ONE question (one question mark only); if you output more than one question, you fail 2. The question must be open-ended (not yes/no) 3. Ask

about their preferences, NOT about factual knowledge 4. Do NOT answer or attempt to fulfill their request 5. Do NOT include any preamble, explanation, or follow-up 6. If there is prior conversation, ask about a DIFFERENT aspect than what was already discussed

The user's request: {original_request}

Your single question:

---

## Human Role-Play Prompt for *User Personalization*

You are roleplaying a person with the following characteristics:

{persona}

You are asking the following question: {original_request}

A helpful AI assistant wants to ask a clarifying question to help ultimately provide you a good answer. Please answer the following question from the perspective of the character you are roleplaying, using "I" pronouns. Make your response sound natural. Crucially, you should never provide an answer to the question. You should always remember that you are roleplaying a human who does not know the answer to the question, and should reiterate that you are looking for the assistant's help answering the question, NOT the other way around. Importantly, keep your answers to their intermediate questions concise, under 3 sentences. Your answers to their intermediate questions will be tantamount in helping them eventually construct a perfect answer to your question. Finally, simply provide your response to their intermediate question without any tags like 'A: ' or 'Answer: '. Below is your conversation history with the assistant.

{conversation_history}
You:

---

## Response-Generation Prompt for *User Personalization*

You are a helpful assistant. A user named {user_name} asked you for help with the following request:

"{original_prompt}"

During your conversation, you asked clarifying questions to better understand their needs. Here is the conversation:

{conversation_text}

Based on this conversation, provide a helpful, personalized response to {user_name}'s original request. Consider the information they shared about their background, preferences, and needs.

Your response should be concise but informative: approximately 100-150 words (1-2 short paragraphs). Give a direct, useful answer that addresses their request while incorporating relevant details from the conversation.

---

## Gold-Response-Generation Prompt for *User Personalization*

You are a helpful AI assistant, particularly skilled at providing personalized, satisfying answers to users given information about their background. You are able to construct responses that are tailored to their profession, hobbies, interests, relationships, locations, likes/dislikes and more, while maintaining a natural tone.

You are answering questions for the following user:

```
{persona}

Answer the question below, tailoring your answer to the user and their characteristics.
Answer directly to the user (i.e., 'you', 'your' pronouns).  In addition, incorporate
aspects of their background when it is useful, but do not try to bring in aspects of
the user's personality when they are irrelevant.  Make sure to keep your answer concise
and organized, but thorough.  Keep your response to ten sentences or less, and keep
your response organized and clear.  Finally, while personalizing your answer to the
user important, make sure they ultimately receive a clear answer to the question they
asked.

{request}
```

**Rating Prompt for *User Personalization***

```
You are an expert in evaluating assistant responses, skilled at analyzing their
alignment with user backgrounds and preferences.

Rate the following assistant response based on how well it aligns with the user's
background and addresses their query.

User Background:  {persona}

User Query:  {request}

Assistant Response:  {response}

Rate the response on a scale of 1-10:  - 1-3:  Poor alignment, generic or irrelevant
to user's background - 4-6:  Moderate alignment, somewhat relevant but could be more
tailored - 7-9:  Good alignment, well-tailored to user's background - 10:  Excellent
alignment, perfectly addresses user's specific needs

Provide brief reasoning (1-2 sentences), then give your score.

You MUST end your response with exactly this format:  Score:  [number from 1-10]
```

**Persona-Generation Prompt for *User Personalization***

```
You are a helpful AI assistant, particularly skilled at writing creative, diverse
personas in the form of characteristics of humans.  These characteristics describe a
user's personality, identity, characteristics, likes and dislikes, social life and
other information.

Take a deep breath.  Please generate exactly one persona describing a user.  IMPORTANT:
Start your persona with "I'm [Name]." where [Name] is the person's first name.  Make
sure you generate personas with a diverse set of information that is interesting,
unpredictable and creative.  Crucially, the persona itself need not exhibit an
appreciation or focus on diversity, although it could; however, what is important is
that you generate an interesting, engaging, creative persona that could serve as a
character in a story with an interesting, complex plot.  The persona should be concise-
less than 10 sentences.  Here are a few examples of personas to help you understand
what sorts of categories of characteristics are important to describe.

{few_shot_examples}
```

## D. Data Curation

The secret words for our 20Qs environment are taken from Google's Trillion Word Corpus (Michel et al., 2011). The 10,000 most common nouns were selected using n-grams (Top English Wordlists) and subsequently filtered using Grok 3 m LLM-as-a-judge to decide which words are suitable and solvable for the game (xAI, 2025). To further ensure that the words of the dataset are appropriate, to collect data for supervised fine-tuning (SFT) and to measure the difficulty of each word, we

played 32 iterations of Twenty Questions with Gemini 2.0 Flash (Gemini Team, 2024). Words that passed the initial filtering but were not solved at least once by Gemini were discarded.

The final dataset consists of 4,678 words. These are randomly split into four disjunctive subsets: 238 words as validation set, 487 as test set and the remainder for model training. Of the 3,953 words remaining for training, the 341 words are chosen for SFT training and 3,612 are reserved for RL training. We perform extensive ablations on the data mixture and choose a set of 1,000 words that are neither too hard nor too difficult for the SFT baseline as final training set. This set is used for all RL training experiments in this study. It is likely that the performance of our method can be further improved by iteratively incorporating secrets at the edge of the agent's capacity into a data curriculum.

# E. SFT Settings

## E.1. SFT Data

As mentioned above in Appendix D, the training set of the BASELINE(SFT) contains gameplay of a distinct set of 341 words. For each of these secrets, we select the successful rollout of Gemini 2.0 Flash with the lowest number of turns, indicating a comparatively higher-quality of questions. If there is a tie between games with equally few turns, we include them all, resulting in a total of 562 sequences.

## E.2. SFT Training

We split the trajectories into individual turns, such that each new question is regarded as its own sample in the training set. The context consisting of the prompts, chat template tokens, previous questions and judge responses are masked, such that we only train on the tokens of the final question for each partial interaction sequence.

We find that this single-turn approach performs better than training with `verl`'s `MultiTurnSFTDataset` on entire sequences at once. Possible explanations are difficulties with the hybrid chat template of `Qwen3` and that it may be beneficial to perform training steps on earlier parts of a sequence before training on the later questions, as this decreases the off-policy gap.

The training is performed using a slightly modified version of `verl`'s `SFTTrainer`. We train for two epochs with cosine decay using a maximum learning rate of `1e-5` and 10% linear warm-up steps. The optimizer is ADAMW with a weight decay of 0.01, $\beta_1$ of 0.9 and $\beta_2$ of 0.95 (Loshchilov & Hutter, 2019). We set the global batch size to 256 with a maximum of 650 tokens per sequence.

# F. GRPO Settings

Our implementation adapts the original Group Relative Policy Optimization (GRPO) algorithm to a multi-turn reinforcement learning setting. We introduce the following modifications to the baseline objective:

- **Token Masking**: Standard GRPO is designed for single-turn prompt-response pairs. In our multi-turn environments, trajectories contain tokens generated by both the agent and the environment simulator. To ensure the policy is only optimized on its own outputs, we apply a mask $M_{i,t}$ (where $i$ denotes the sample and $t$ the turn) to exclude environment-generated tokens from the loss calculation.

- **KL Divergence Omission**: Contrary to the original formulation, we found that including a Kullback–Leibler (KL) divergence penalty did not yield significant performance gains in our setting. Consequently, we omit the KL term from the final loss function (see Figure 17 for an empirical comparison).

- **Asymmetric Clipping**: We employ an asymmetric clipping range to allow for larger policy updates in the positive direction while maintaining stability, as introduced by DAPO (Yu et al., 2026). Specifically, we set the upper clipping bound to $\epsilon_{\text{high}} = 0.28$ and the lower bound to $\epsilon_{\text{low}} = 0.2$.

- **Per-Turn Advantage Estimation**: As detailed in Section 3.2, we calculate advantages at a per-turn granularity rather than over the entire trajectory. This approach reduces variance and provides a more precise credit assignment for specific information-seeking actions.

### F.1. StarPO Ablations

To ensure that CIA is compared fairly, we performed multiple ablations on the GRPO baseline (StarPO). These focus on the aggregation method of the loss and learning rate. Especially the latter was shown to be relevant in our specific setting, with `sequence-mean-token-mean` aggregation being necessary for optimal training. These baselines use the hyperparameter combination that was found to be ideal in the ablations and the default, sequence-wise GRPO-objective. Unless otherwise noted, all other parameters are exactly the same as for our method, including using LoRA and the rewards for the final, verifiable outcome and for efficient exploration, as introduced in Section A.1.

*Table 5.* **Aggregation methods and learning rates.** Ablation results of different aggregation methods and learning rates for the GRPO training of `Qwen3-1.7B` (STARPO-1.7B). The best result is shown in bold.

| Aggregation | LR = $1 \times 10^{-5}$ | LR = $3 \times 10^{-5}$ |
|---|---|---|
| seq-mean-token-mean | **16.54%** | 15.85% |
| token-mean | 12.24% | 12.50% |

*Table 6.* **Aggregation methods and learning rates.** Ablation results of different aggregation methods and learning rates for the GRPO training of `Qwen3-4B` (STARPO-4B). The best result is shown in bold.

| Aggregation | LR = $1 \times 10^{-5}$ | LR = $3 \times 10^{-5}$ |
|---|---|---|
| seq-mean-token-mean | 20.21% | **24.36%** |
| token-mean | 21.07% | 21.54% |

