# OpenReview forum: "Intrinsic Credit Assignment for Long Horizon Interaction"
_ICML.cc/2026/Conference — ICML 2026 regular_

### Official Review · Reviewer_K4z8 · 2026-03-01

**Soundness:** 3
**Presentation:** 4
**Significance:** 3
**Originality:** 3
**Overall Recommendation:** 4
**Confidence:** 4

**Summary:**

This paper proposes a ΔBelief-RL method for long-horizon interactive environments with sparse rewards. It constructs a dense, turn-level reward by measuring the per-turn increase in belief (log-probability) over the ground-truth target, and combines it with trajectory-level outcome for GRPO training.

**Compliance With Llm Reviewing Policy:**

Affirmed.

**Final Justification:**

The author did solve most of my problems, and I have increased my score accordingly.

**Key Questions For Authors:**

1. The paper’s core signal ($\Delta$Belief as turn-level log-probability improvement on the ground-truth target) appears closely related in spirit to IGPO[1] and TIPS[2] turn-level information-gain reward, which also uses incremental increases in ground-truth log-likelihood. Please discuss the key conceptual differences (e.g., target definitions, signal's roles) and why those differences matter.
2. Can $\Delta$Belief-RL be evaluated or at least discussed on IGPO-style[1] QA benchmarks? A short experiment or discussion would help clarify novelty and scope.

[1] Information Gain-based Policy Optimization: A Simple and Effective Approach for Multi-Turn LLM Agents

[2] TIPS: Turn-level Information-Potential Reward Shaping for Search-Augmented LLMs

**Limitations:**

yes

**Strengths And Weaknesses:**

### Strengths:
1. In the long-horizon environment, reward shaping for sparse reward is a crucial problem, and ΔBelief-RL constructs a dense, turn-level reward by measuring the per-turn increase in belief (log-probability) over the ground-truth target.
2. Extensive experiments on several datasets validate ΔBelief-RL's strong performance over baseline and STARPO.
3. The paper is very rich in content. Extended discussions on belief's reflection on interactive progress, comparisons of different reward normalizations, ablations of reward compositions, and OOD generalization make the results more convincing.

### Weaknesses:
1. I agree with the limitations discussed in the paper. $\Delta$Belief-RL seems effective only for information-seeking tasks which have clear answers for each task, but cannot be applied to more general environments.
2. The paper draws several conclusions based primarily on 1.7B and 4B models. This seems insufficient to support broader claims about scaling behavior or generality. Adding a larger model (e.g., 8B) and reporting the same comparisons would strengthen the conclusions.
3. The method requires computing/collecting per-turn belief signals during training. It is unclear how this affects sampling throughput and wall-clock training time relative to trajectory-level GRPO. A clear cost breakdown would make the “efficiency” claims more convincing.
---
typo:
The manuscript contains three missing cross-references: Line 2289, Line 2812, Line 2828.

---

> ### Author Rebuttal · Authors · 2026-03-31
>
> We thank you for the thoughtful and constructive feedback, and for recognizing that we have performed “extensive experiments on several datasets” to validate our method, that overall the “paper is very rich in content” and that we address an important issue "reward shaping for sparse reward is a crucial problem". We now address the concerns raised:
>
> Q.1./Q.2.  “Conceptual differences from IGPO[1] and TIPS[2] (target definitions, signal’s roles)”; “Can Belief-RL be evaluated or at least discussed on IGPO-style[1] QA benchmarks?”
>
> We did not include a ΔBelief-RL comparison with IGPO as it was concurrent to our work. However, we now have implemented IGPO training on our benchmark and compared it to ΔBelief-RL and present results here: https://imgur.com/a/nbUyZyL. We find that IGPO-style training is **less effective** for our **question-asking, information-seeking** setting. A likely reason being the different credit attribution choices across the methodologies: we propose a local per turn attribution, whilst IGPO takes into account all future discounted rewards.
>
> For completeness, we also ablate IGPO intrinsic reward **z-normalization** in our framework to verify whether it can be directly applied in our training pipeline. In particular, we compare our original ReLU shaping with the trajectory level z-norm as described in IGPO, with results here: https://imgur.com/a/nbUyZyL (red). The figure shows that z-norm is not beneficial in our training setup, suggesting that z-norm distorts the local directionality of the signal.
>
> Taken together, these results suggest that there are important methodological differences that pose ΔBelief-RL better suited for our long-horizon interaction tasks. Specifically, ΔBelief-RL preserves **turn-local credit assignment**, whereas IGPO via discounting mixes local signal with future intermediate information gain and TIPS aggregates teacher's probability of any acceptable answer. However, we are excited that the same high-level ideas have worked for the concurrent work on the domain of tool/search-augmented question answering and will cite them in revised draft, as they together with our work indicate that broader direction of intrinsic intermediate rewards enable many new research directions.
>
> W.1. “Belief-RL seems effective only for information-seeking tasks which have clear answers for each task, but cannot be applied to more general environments.”
>
> We show strong generalization to tasks where the answer of the environment is a complex description (Sec.6.2. Murder Mystery; Sec.6.3 User Personalization). This indicates that the induced information seeking behavior by ΔBelief-RL is not limited to environments with binary answers. More importantly, it highlights that the skill elicited by ΔBelief-RL is the ability to ask informative open-ended questions. Our training framework can be applied to any task where log probabilities are meaningful. Note, that our method first aggregates the token log probabilities of the target phrase, and, importantly, computes the logprob differences of the same phrase across turns. Thus, the method itself and the learned agent is not limited to tasks with binary answers.
>
> W.2. “Conclusions are based primarily on 1.7B and 4B models”
>
> Based on your suggestion we are currently training a Qwen3-8B model and we will include it in the revised version. We note that  we emphasize scaling w.r.t test-time interaction i.e. long horizon (Results in Fig.9). Our results showcase that the trained information seeking behavior generalizes surprisingly well beyond the training length cut-off.
>
> W.3. “cost breakdown”
>
> Despite the additional forward pass, CIA reaches GRPO's peak performance 28% faster in wall-clock time (https://imgur.com/a/iZHxVQo), thanks to improved sample efficiency. The per-step overhead is modest, with +10% for Qwen3-1.7B and +15% for Qwen3-4B (https://imgur.com/a/x329GvG). This is achieved through a custom attention mask (similar to Tree Attention [1]) that computes the Delta-Belief for all turns in a single forward pass instead of one pass per turn. We will add these timing results to the revised paper.
>
> [1] Medusa: Simple LLM Inference Acceleration Framework with Multiple Decoding Heads
>
> We hope that we have addressed all major concerns with additional ablations and clarifications. We appreciate the high scores (Soundness: 3, Presentation: 4, Significance: 3, Originality: 3) and would be grateful if the reviewer would consider raising their score.

---

> > ### Author Rebuttal · Reviewer_K4z8 · 2026-04-03
> >
> > The author did solve most of my problems, and I have increased my score accordingly. However, I still think the applicability of the method is somewhat limited. It seems primarily designed for long-horizon information-seeking tasks with a clear ground-truth target or reference answer, where belief updates can be explicitly tracked. Its extension to more general environments—especially open-ended, multi-solution, or weakly specified tasks without a well-defined target—remains unclear, and I would encourage the authors to state this boundary more explicitly.

---

### Official Review · Reviewer_WZPE · 2026-03-11

**Soundness:** 3
**Presentation:** 3
**Significance:** 2
**Originality:** 2
**Overall Recommendation:** 5
**Confidence:** 3

**Summary:**

The authors present a novel method to densify the reward signal in long-horizon tasks. They use the model’s internal knowledge to measure progress towards the goal through the notion of belief.

**Compliance With Llm Reviewing Policy:**

Affirmed.

**Final Justification:**

I find the paper well-written, with an intuitive and original idea. The authors addressed my concerns effectively, and I have increased the score accordingly.

**Key Questions For Authors:**

- Do you have Oracle results for “Murder Mystery” and “Guess My City” to compare the zero-shot performance? A good oracle would be to train CIA and other baselines on these environments as well.
- How robust is CIA when interacting with a weaker simulator? Can you provide an experiment where the judge is a smaller LM?

**Limitations:**

yes

**Strengths And Weaknesses:**

**Strengths:**

- Novel dense reward mechanism for more efficient training in long-horizon tasks. (Originality)
- Significant performance improvement across all experiments. (Significance)
- The method exhibits interesting zero-shot performance on OOD tasks. (Significance)

**Weaknesses:**
- The target solution needs to be known beforehand to calculate the belief, making it a very limited framework. There are setups where the final solution depends on interactions and can only be formulated later. (Significance/Soundness)
- Computational overhead: CIA requires an additional forward pass each iteration to calculate the belief. Can you provide a table showcasing this? (Soundness/Presentation)

---

> ### Author Rebuttal · Authors · 2026-03-30
>
> We thank you for your helpful feedback and for highlighting the novelty of our dense reward mechanism, “significant performance improvement across all experiments”, and the “interesting zero-shot performance on OOD tasks” shown by our method.
>
> We address your remaining questions below.
>
>
> **W1:** "The target solution needs to be known beforehand to calculate the belief, making it a very limited framework. There are setups where the final solution depends on interactions and can only be formulated later."
>
> We thank you for this point and acknowledge we could have discussed the scope more clearly. We would like to clarify a common misunderstanding: ΔBelief-RL does not require the target to be known before the interaction. The reward can be computed post-hoc on stored trajectories, meaning the ground truth only needs to be available at reward-computation time. This naturally extends to settings where (1) the target emerges during the conversation, or (2) the best response across a group of generations serves as the shared target.
>
> The core requirement is that progress toward a target can be measured via the model's belief. This condition is satisfied in any verifiable-outcome, multi-turn setting. These are precisely the domains where outcome-based RLVR (GRPO and variants) has proven most impactful, yet suffers from lacking credit assignment across turns. ΔBelief addresses this gap directly. We have expanded this discussion in the revised manuscript.
>
>
> **Q1:** "Do you have Oracle results for “Murder Mystery” and “Guess My City” to compare the zero-shot performance? A good oracle would be to train CIA and other baselines on these environments as well."
>
> We agree that this is a good suggestion. Training CIA and baselines directly on the training sets of Murder Mystery and Guess My City would provide a useful upper bound for contextualizing our zero-shot results. We are running these experiments and will add this to the final draft.
>
>
> **Q2:** "How robust is CIA when interacting with a weaker simulator? Can you provide an experiment where the judge is a smaller LM?"
>
> We agree that robustness to different simulators is important and that using a smaller one can positively impact the training time. Some experiments where CIA interacts with other judges at inference time can be found in Table 2. We expand those results with experiments using a smaller Qwen3-4B judge below:
>
> | **Judge** | **CIA-4B** |
> | --- | --- |
> | Qwen3-4B (**new**) | 31.00% |
> | Qwen3-14B (train) | 35.65% |
> | Qwen3-235B-A22B-Instr. | 31.18% |
> | DeepSeek-v3.2 | 37.64% |
>
> We observe that our agent's performance remains robust, comparable to Qwen3-235B-A22B-Instruct. This shows that CIA generalizes across judges not seen during training.
>
> **W2:** "Computational overhead: CIA requires an additional forward pass each iteration to calculate the belief. Can you provide a table showcasing this?"
>
> Despite the additional forward pass, CIA reaches GRPO's peak performance 28% faster in wall-clock time (https://imgur.com/a/iZHxVQo), thanks to improved sample efficiency. The per-step overhead is modest, with +10% for Qwen3-1.7B and +15% for Qwen3-4B (https://imgur.com/a/x329GvG). This is achieved through a custom attention mask (similar to Tree Attention [1]) that computes the ΔBelief for all turns in a single forward pass instead of one pass per turn. We will add these timing results to the revised paper.
>
> We hope we have addressed the major concerns of the reviewer, and are happy to answer any further questions/concerns. We look forward to a fruitful reviewer-author discussion phase.
>
> [1] Medusa: Simple LLM Inference Acceleration Framework with Multiple Decoding Heads

---

> > ### Author Rebuttal · Reviewer_WZPE · 2026-04-01
> >
> > This resolves my issues. I have increased the significance score and the overall recommendation accordingly. Very good paper!

---

### Official Review · Reviewer_ErKz · 2026-03-12

**Soundness:** 3
**Presentation:** 3
**Significance:** 3
**Originality:** 3
**Overall Recommendation:** 4
**Confidence:** 4

**Summary:**

The paper proposes $\Delta$Belief-RL, a reinforcement learning framework for long-horizon language-agent interaction that uses the model’s own belief change toward the target as a dense intrinsic reward for intermediate credit assignment. The method avoids a separately trained critic by deriving turn-level training signals directly from changes in the model’s elicited belief over the correct answer. Experiments on 20 Questions indicate improvements over baselines. Overall, the idea is novel and appealing, though there is still room to further strengthen the empirical validation and analysis.

**Compliance With Llm Reviewing Policy:**

Affirmed.

**Final Justification:**

I view the paper positively, with its strengths outweighing its weaknesses, and the rebuttal helpfully resolved my main confusion. However, the clarification was not sufficient to change my evaluation, so I keep my original score unchanged.

**Key Questions For Authors:**

I have no additional questions beyond those already raised in the weaknesses section.

**Limitations:**

yes

**Strengths And Weaknesses:**

Strengths:

1. $\Delta$Belief-RL does not rely on a critic, but instead uses the model’s own internal belief signal. This is a simple and elegant design choice, and the experiments provide evidence for its effectiveness.

2. The empirical results are encouraging, and on the 20 Questions benchmark the proposed method shows clear improvements over the baselines.

3. The paper is well written and easy to follow, with a clear and well-motivated presentation of the core idea.

4. The paper takes the extra step of validating the proposed $\Delta$Belief signal before incorporating it into RL training, which strengthens the overall methodological rigor.

Weaknesses:

1. While the belief-based design is interesting, the proposed belief construction is not yet fully convincing. In particular, the method relies on $b_t = p_\theta(y \mid h_t, e)$ as a proxy for the agent's internal belief, where the measured quantity depends on both the elicitation prompt $e$ and the current interaction history $h_t$. Since language model probabilities can be sensitive to prompt phrasing and surface-form biases, they do not necessarily correspond to a well-calibrated posterior update over the target concept. Therefore, it is unclear to what extent $\Delta$Belief reflects genuine progress in uncertainty reduction, rather than prompt-dependent scoring effects.

2. Since the main experiments are centered on 20 Questions, a setting particularly well matched to the $\Delta$Belief design, it remains unclear how well the method generalizes to less structured interaction settings with ambiguous or noisy feedback.

3. The current framework appears to rely on access to the correct target during training, and may therefore be better suited to tasks with verifiable and uniquely specified answers than to more open-ended interaction settings.

4. Although the supplement is extensive, much of it is devoted to qualitative examples and auxiliary prompt content rather than stronger quantitative validation, so its length does not fully translate into stronger empirical support. Moreover, the lack of a code release also limits reproducibility, and a complete implementation would likely be more useful than a lengthy supplement.

---

> ### Author Rebuttal · Authors · 2026-03-31
>
> We appreciate that you found our core idea of leveraging the model's internal belief signal to be a "*simple and elegant design choice*" and our overall framework "*novel and appealing*". We also thank you for recognizing our methodological rigor in validating the belief signal prior to RL training, alongside the "*clear improvements*" demonstrated in our empirical results. Below, we address the specific points you raised in detail:
>
> **W1.** “*the proposed belief construction is not yet fully convincing* [..] *it is unclear to what extent Belief reflects genuine progress in uncertainty reduction, rather than prompt-dependent scoring effects*”.
>
> In Section 3.1, we empirically validate that ΔBelief correlates with trajectory outcomes and improves performance when used as sampling guidance, providing direct evidence that it reflects genuine progress rather than scoring artifacts. By design, our formulation measures the difference in log-probabilities between consecutive turns. Static biases such as surface-form effects and concept-specific priors cancel out, isolating the information gain provided by the latest observation.
>
> We additionally conducted a prompt sensitivity analysis across multiple elicitation formats (varying syntactic framing and tokenization) and model sizes; see the results presented in the [following table](https://imgur.com/a/7yGJXK7). We kept the final elicitation format minimal (e.g., "Q: [...]\nA: [...]") to further reduce confounding factors. We will include this analysis in the appendix of the camera-ready version.
>
> **W2.** / **W3.** “*it remains unclear how well the method generalizes to less structured interaction settings with ambiguous or noisy feedback*” and “*better suited to tasks with verifiable and uniquely specified answers than to more open-ended interaction settings*”.
>
> We acknowledge we could have discussed the scope and applicability of our method more clearly. We would like to clarify two points:
>
> Generalization at test time: Overcoming the structured-environment limitation at test time was a primary motivation for our work. We prioritized evaluating generalization to less structured, open-ended settings in Section 6. Models trained with ΔBelief-RL on verifiable data exhibit strongly enhanced information-seeking capabilities in open-ended downstream tasks, evaluated OOD (User Personalization, Customer Service). This suggests that ΔBelief-RL develops fundamental information-seeking abilities that transfer across environment dynamics and input structures.
>
> Scope at training time: While our current experiments use verifiable targets, ΔBelief does not require the target to be known before the interaction. The reward is computed upon completion on stored trajectories, so the ground truth only needs to be available at reward-computation time. This naturally extends to settings where the correct answer emerges during the conversation or where the best final solution in a group of trajectories serves as a shared target. ΔBelief addresses the lack of credit assignment of traditional outcome-based methods in these open, multi-turn settings.
>
> Regarding noise, we note that the 20-Questions environment is quite noisy: binary yes/no answers to very open-ended questions generated by the model trained a major source of ambiguity (not an alleviation), and reliance on an LLM judge as the transition function introduces further stochasticity. Our method is robust under these conditions as shown by the OOD generalization of Section 6 and inference-time changes of the judge in Table 2.
>
> **W4.** “*supplement* [..] *does not fully translate into stronger empirical support.* [..] *lack of a code release*”
>
> We appreciate the feedback regarding the appended supplements. To strengthen the supplementary material, we have added details about our efficient ΔBelief reward implementation, timing analyses, and ablations of the elicitation. Further suggestions for additional quantitative analyses are appreciated. We are committed to open science and have published our source code along with training and evaluation scripts in the following anonymized repository: https://github.com/IntrinsicCreditAssignment/delta-belief-rl
>
> We hope we have addressed your major concerns and are happy to answer any further questions reviewer-author discussion.

---

> > ### Author Rebuttal · Reviewer_ErKz · 2026-04-01
> >
> > I appreciate the authors’ clarification and the added elicitation-prompt ablation, which helps address this concern. My overall assessment remains unchanged, and I keep my original score at 4.

---

### Official Review · Reviewer_hVJN · 2026-03-14

**Soundness:** 3
**Presentation:** 3
**Significance:** 2
**Originality:** 2
**Overall Recommendation:** 3
**Confidence:** 3

**Summary:**

This paper presents ∆Belief-RL, a reinforcement learning framework that leverages a language model's own intrinsic beliefs to reward intermediate progress for training LLM in multi-turn, partial information environments.

**Compliance With Llm Reviewing Policy:**

Affirmed.

**Key Questions For Authors:**

- How does the theoretical formulation of the  ∆Belief reward fundamentally differ from the stepwise confidence gain proposed in related works like PACR?

- What is the main reason behind modifying the GRPO? Did these modification made results better than standard GRPO?

**Limitations:**

yes

**Strengths And Weaknesses:**

#### Strengths

- This paper is generally well written and easy to follow with a simple and representative experiment design.

- Utilizing the model's internal log-probabilities directly avoids the cost of training and deploying a separate process reward model, and the results demonstrate that this signal can be beneficial.

- The trained models using \delta-belief-rl enable the model with higher and generalized capabilities of long-horizon interaction over OOD tasks like murder mystery, indicating the successful learning of information-seeking techniques.

#### Weakness

- The core mechanism seems to be a combination highly similar  of several recent work over multi-turn rl for LLM and token-level log-probabilities as signal. This paper is then essentially an application level combination of them without further novel design.

- The framework requires an elicitation prompt to measure the probability of the target. While this leads to policy that generalizes, the training framework itself is then limited to environments where a single, easily elicitable ground-truth noun is known.

- The primary training and evaluation rely on the 20 Questions environment with immediate feedback. While this task requires interaction, it is actually highly constrained with direct binary-response that does not adequately reflect the complexity and openess or the necessity of multi-step reasoning required for most infromation seeking tasks and agents.

- It seems in Fig.1 the GRPO results are not further explained.  The GRPO success rate peaks early and then severely drops, this could potentially due to the use of BF16 or the omission of KL term? At least the GRPO with KL and without should both present in the results comparison, comparing with the best results within GRPO with other modifications and the original GRPO.

---

> ### Author Rebuttal · Authors · 2026-03-30
>
> We thank you for the thoughtful feedback and for highlighting that our method results in a “successful learning of information-seeking techniques”, that "the results demonstrate that this signal can be beneficial" and avoids "the cost of training and deploying a separate process reward model". We address your concerns below:
>
> Q.1./W.1.: “theoretical formulation difference from PACR step-wise confidence gain”; “application level combination without further novel design”
>
> We discuss PACR in our related works section, please find here an extended comparison. From a theoretical formulation PACR step-wise confidence gain measures the confidence in a known correct final answer along a reasoning trace, i.e. reasoning-trace shaping. On the contrary, our work measures posterior belief update over a hidden target induced by an action-observation turn in a partially observed interaction. Furthermore, we differ from PACR in the following 2 ways: (1) reward construction; (2) signal combination. First, PACR-Dense takes the stepwise confidence gains and forms a discounted future reward. We, however, combine outcome reward, intrinsic reward and penalties first at the reward level per turn creating a local reward signal. Second, PACR-Dense applies min-max normalisation to create step-wise advantages and combines with standard GRPO at the advantage level, whilst ΔBelief-RL computes turn-wise GRPO advantage from the combined reward. We chose to combine signals at the reward space first, as combining signals after separate normalisation can distort their relative scale and semantics (see additional results on z-norm ablation for Q.2. reviewer K4z8). In addition, we ablated the min-max normalisation by PACR in our training framework, and found that simple ReLU works best (Fig. 8), indicating that the signal’s relative magnitude matters. Thus PACR is a method for shaping single-trajectory reasoning, whereas our formulation targets credit assignment for information-seeking actions in long-horizon interaction.
>
> Q.2.: “GRPO modification reason”
>
> We modify GRPO because standard implementation aggregates reward at the trajectory level, collapsing turn-level credit assignment into a single signal. While this is fine for sequence-level penalties, it distorts credit assignment, core to the  ΔBelief reward, which is designed to measure which turn/action caused progress. We therefore use a turn-wise GRPO formulation. Empirically, this performs better than standard GRPO in our setting (https://imgur.com/a/QUH85c2).
>
> W.4.: “Fig.1. clarification; GRPO results are not further explained.”
>
> We would like to clarify this point. All RL runs were trained with the same precision, so the drop in GRPO performance in Fig. 1 is not due to differing precision settings. Additionally, we evaluate KL vs. no-KL for both standard GRPO and ΔBelief-RL, and observe no benefit from KL in either case (https://imgur.com/mbI79ZF; Fig.13). For this reason, we omit KL in the final method. We now additionally compare several loss aggregation schemes for standard GRPO and adopt the strongest-performing variant as our baseline, to ensure comparison against the best GRPO-style training setup we tested (https://imgur.com/F8sRmVa).
>
> W.2: “limited to environments where  a single elicitable gt noun is known”
>
> The framework can be applied to any task where log probabilities are meaningful. Note that our method first aggregates the token log probabilites of the target pharse, and, importantly, computes the **logprob differences** of the same phrase across turns, thus, the absolute value of a single concept is not relevant. We want to highlight that models post-RL trained with ΔBelief-RL exhibit strongly enhanced information-seeking capabilities on tasks where the environment responses are complex descriptions (Sec.6. Customer Service, User Personalisation, Muder Mystery) compared to standard GRPO. Thus, showing clear evidence of behavior generalisation beyond single noun concepts.
>
> W.3: “direct binary-response that does not adequately reflect the complexity and openess of [..] multi-step reasoning required for information seeking tasks”
>
> We show extensive evaluations on other tasks where the environment responses are open-ended. In all of these OOD tasks our framework shows improved peformance (Sec.6.2-6.3), indicating that the learned behavior generalises beyond direct binary-response tasks. Moreover, this highlights that our method results in a general information seeking skill of asking informative open-ended questions compared to standard GRPO.
>
> We hope that we have addressed all major concerns and are happy to answer any further questions. We look forward to a fruitful reviewer-author discussion phase.

---

> > ### Author Rebuttal · Reviewer_hVJN · 2026-04-03
> >
> > I thank the authors for the response. I maintain my ratings.

---

> > > ### Author Response · Authors · 2026-04-08
> > >
> > > Dear Reviewer hVJN,
> > >
> > > Thank you for your constructive feedback, which has helped us strengthen our paper. We have conducted additional experiments to address your main concerns: clarified that our framework can be applied to any task where log probabilities are meaningful (as it only depends on the difference between logprob values), performed additional ablations on the effect of KL term with GRPO, compared with PACR normalisation (and similarly IGPO in rebuttal to K4z8), showed generalization to other open-ended environments (section 6.2-6.3). These experiments strongly support our original findings and hypotheses. As you have indicated that all your concerns are fully resolved and have not followed up by any additional questions, we would be very grateful if you would consider raising your score in support of further progress in this research area. We are happy to answer any further questions.

---

### Decision · Program_Chairs · 2026-04-30

**Decision:**

Accept (regular)

**Comment:**

This paper provides a simple idea for generating dense rewards based on a model's own log-probabilities as it gathers information. The idea can work for any setting where a ground truth "correct" answer is available post-hoc when the rewards are computed. An additional forward pass is necessary, but it can be done efficiently with a special purpose mask design. The paper provides a rigorous empirical evaluation of this idea, beyond just RL training performance. Some reviewers maintain concerns about comparisons to related work, but these were largely addressed.